# Porcupine Neural Networks:
# Approximating Neural Network Landscapes

**Soheil Feizi**
Department of Computer Science
University of Maryland, College Park
sfeizi@cs.umd.edu

**Hamid Javadi**
Department of Electrical and Computer Engineering
Rice University
hrhakim@rice.edu

**Jesse Zhang**
Department of Electrical Engineering
Stanford University
jessez@stanford.edu

**David Tse**
Department of Electrical Engineering
Stanford University
dntse@stanford.edu

## Abstract

Neural networks have been used prominently in several machine learning and statistics applications. In general, the underlying optimization of neural networks is non-convex which makes analyzing their performance challenging. In this paper, we take another approach to this problem by constraining the network such that the corresponding optimization landscape has good theoretical properties without significantly compromising performance. In particular, for two-layer neural networks we introduce Porcupine Neural Networks (PNNs) whose weight vectors are constrained to lie over a finite set of lines. We show that most local optima of PNN optimizations are global while we have a characterization of regions where bad local optimizers may exist. Moreover, our theoretical and empirical results suggest that an unconstrained neural network can be approximated using a polynomially-large PNN.

## 1 Introduction

Neural networks have been used in several machine learning and statistical inference problems including regression and classification tasks. Some successful applications of neural networks and deep learning include speech recognition [1], natural language processing [2], and image classification [3]. The underlying neural network optimization is non-convex in general which makes its training NP-complete even for small networks [4]. In practice, however, different variants of local search methods such as the gradient descent algorithm show excellent performance. Understanding the reason behind the success of such local search methods is still an open problem in the general case.

There have been several recent works in the theoretical literature aiming to study risk landscapes of neural networks and deep learning under various modeling assumptions. For example, references [5, 6, 7, 8] have studied the convergence of the local search algorithms for the neural network optimization with zero hidden neurons and a single output. Other works have studied the risk landscapes of neural network optimizations for more complex structures under various model assumptions [9, 10, 11, 12, 13, 14, 15, 16, 17, 18, 19]. In particular, references [13, 14, 15] consider a two-layer neural network with Gaussian inputs under a matched (realizable) model where the output is generated from a network with planted weights. Over-parameterized networks where the number of parameters are larger than the number of training samples have been studied in [20, 21]. In addition, reference [22] makes the argument (through a visualization scheme) that the loss landscape of reduced-capacity networks such as resnets has fewer bad locals than that of unconstrained ones. Further, [23] presents

conditions for a local minimum to be global in the case of having regularization on the network weights.

We review these prior work in SM [1] Section 2. In this paper, we study a key question: can an unconstrained neural network be approximated with a constrained one whose optimization landscape has good theoretical properties? For two-layer neural networks, we provide an affirmative answer to this question by introducing a family of constrained neural networks which we refer to as *Porcupine Neural Networks (PNNs)* (Figure 1-a). In PNNs, incoming weight vectors to hidden neurons are constrained to lie on a fixed set of lines. For example, a neural network with multiple inputs and multiple neurons where each neuron is connected to one input is a PNN since input weight vectors to neurons lie on lines parallel to the standard axes (SM Section 3).

Designing new objective functions with good landscape properties for non-convex problems has been explored previously in [19, 15, 24]. Our work provides an alternative procedure to obtain a new objective function for the underlying non-convex optimization. Our framework can be extended naturally to deeper networks as well even though in our analysis in this paper, we focus on two-layer networks. Moreover, a PNN can be viewed as a neural network whose feature vectors (i.e., input weight vectors to neurons) are fixed up to scalings due to the PNN optimization. This view can relate a random PNN (i.e., a PNN whose lines are random) to the application of random features in kernel machines [25, 26, 27, 28, 29, 30, 6]. Although our results in Sections 4 and 5 are for general PNNs, we study them for random PNNs in Section 6 as well.

We analyze population risk landscapes of two-layer PNNs with jointly Gaussian inputs and rectified linear unit (ReLU) activation functions at hidden neurons. We show that under some modeling assumptions, most local optima of PNN optimizations are also global optimizers. Moreover, we characterize the parameter regions where bad local optima (i.e., local optimizers that are not global) may exist.

Next, we study whether one can approximate an unconstrained (fully-connected) two-layer neural network function with a PNN whose number of neurons is polynomially-large in dimension. Our empirical results offer an affirmative answer to this question. For example, suppose the output data is generated using an unconstrained two-layer neural network with $d = 15$ inputs and $k^* = 20$ hidden neurons. Using this data, we train a random two-layer PNN with $k$ hidden neurons. We evaluate the PNN approximation error as the mean-squared error (MSE) normalized by the $L_2$ norm of the output samples in a two-fold cross validation setup. As depicted in Figure 1-b, by increasing the number of PNN hidden neurons $k$, the PNN approximation error decreases. Notably, to obtain a relatively small approximation error, PNN's number of hidden neurons does not need to be exponentially large in dimension. We explain details of this experiment in Section 7.

In Section 6, we study a characterization of the PNN approximation error with respect to the input dimension and the complexity of the unconstrained neural network function. We show that under some modeling assumptions, the PNN approximation error can be bounded by the spectral norm of the generalized Schur complement of a kernel matrix. We analyze this bound for random PNNs in the high-dimensional regime when the ground-truth data is generated using an unconstrained neural network with random weights. For the case where the dimension of input and the number of hidden neurons increase with the same rate, we compute the asymptotic limit. Finally, in Section 8, we discuss how the proposed PNN framework can potentially be used to *explain* the success of local search methods such as gradient descent in solving the unconstrained neural network optimization.

In summary, PNNs provide three main advantageous. The first is that its loss landscape has nice properties with provably few bad locals. Thus, we can have some guarantees on the performance of gradient descent. Second, its approximation power is good, i.e. our experimental and theoretical results suggest that one can approximate unconstrained networks arbitrarily closely with polynomially-large PNNs. Third, PNN uses fewer parameters than unconstrained neural networks, thus leading to better generalization and also benefits in terms of storage. Finally, we discuss using the PNN framework in general deep neural networks in Section 8.

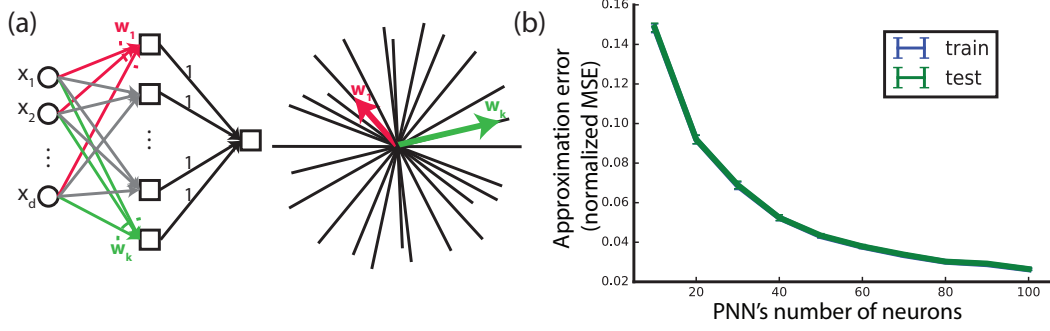

Figure 1: (a) A two-layer Porcupine Neural Network (PNN). In PNN, incoming weight vectors to neurons are constrained to lie over a fixed set of lines in a $d$-dimensional space. (b) Approximations of an unconstrained two-layer neural network with $d = 15$ inputs and $k^* = 20$ hidden neurons using random two-layer PNNs.

## 2  Unconstrained Neural Networks

Consider a two-layer neural network with $k$ neurons where the input is in $\mathbb{R}^d$ (Figure 1-a). The weight vector from the input to the $i$-th neuron is denoted by $\mathbf{w}_i \in \mathbb{R}^d$. For simplicity, we assume that second layer weights are equal to one. Let

$$h(\mathbf{x}; \mathbf{W}) := \sum_{i=1}^{k} \phi\left(\mathbf{w}_i^t \mathbf{x}\right), \tag{1}$$

where $\mathbf{x} = (x_1, ..., x_d)^t$ and $\mathbf{W} := (\mathbf{w}_1, \mathbf{w}_2, ..., \mathbf{w}_k) \in \mathcal{W} \subseteq \mathbb{R}^{d \times k}$. The activation function at each neuron is assumed to be $\phi(z) := \text{ReLU}(z) = \max(z, 0)$.

Consider $\mathcal{F}$, the set of all functions $f : \mathbb{R}^d \to \mathbb{R}$ where $f$ can be realized with a neural network described in (1). In other words,

$$\mathcal{F} := \left\{ f : \mathbb{R}^d \to \mathbb{R}; \;\; \exists \mathbf{W} \in \mathcal{W}, \;\; f(\mathbf{x}) = h(\mathbf{x}; \mathbf{W}), \;\; \forall \mathbf{x} \in \mathbb{R}^d \right\}. \tag{2}$$

In a fully connected neural network structure, $\mathcal{W} = \mathbb{R}^{d \times k}$. We refer to this case as the unconstrained neural network. Note that particular network architectures can impose constraints on $\mathcal{W}$.

Let $\mathbf{x} \sim \mathcal{N}(0, \mathbf{I})$. We consider the population risk defined as the mean squared error (MSE):

$$L(\mathbf{W}) := \mathbb{E}\left[ (h(\mathbf{x}; \mathbf{W}) - y)^2 \right], \tag{3}$$

where $y$ is the output variable. If $y$ is generated by a neural network with the same architecture described by (1), we have $y = h(\mathbf{x}; \mathbf{W}_{true})$.

Understanding the population risk function is an important step towards characterizing the empirical risk landscape [5]. In this paper, for simplicity, we only focus on the population risk. The neural network optimization can be written as follows:

$$\min_{\mathbf{W}} \quad L(\mathbf{W}) \tag{4}$$
$$\mathbf{W} \in \mathcal{W}.$$

Let $\mathbf{W}^*$ be a global optimum of this optimization. $L(\mathbf{W}^*) = 0$ means that $y$ can be generated by a neural network with the same architecture (i.e., $\mathbf{W}_{true}$ is a global optimum.). We refer to this case as *matched*. Moreover, we refer to the case of $L(\mathbf{W}^*) > 0$ as *mismatched*. Optimization (4) in general is non-convex owing to nonlinear activation functions in neurons.

## 3  Porcupine Neural Networks

Characterizing the landscape of the objective function of optimization (4) is challenging in general. In this paper, we consider a constrained version of this optimization where weight vectors belong to a

finite set of lines in a $d$-dimensional space (Figure 1-a). This constraint may arise either from the neural network architecture or can be imposed by design.

Mathematically, let $\mathcal{L} = \{L_1, ..., L_r\}$ be a set of lines in a $d$-dimensional space. Let $\mathcal{G}_i$ be the set of neurons whose incoming weight vectors lie over the line $L_i$. Therefore, we have $\mathcal{G}_1 \cup ... \cup \mathcal{G}_r = \{1, ..., k\}$. Moreover, we assume $\mathcal{G}_i \neq \emptyset$ for $1 \leq i \leq r$; otherwise that line can be removed from the set $\mathcal{L}$. For every $j \in \mathcal{G}_i$, we define the function $g(.)$ such that $g(j) = i$.

For a given set $\mathcal{L}$ and a neuron-to-line mapping $\mathcal{G}$, we define $\mathcal{F}_{\mathcal{L},\mathcal{G}} \subseteq \mathcal{F}$ as the set of all functions that can be realized with a neural network (1) where $\mathbf{w}_i$ lies over the line $L_{g(i)}$. Namely,

$$\mathcal{F}_{\mathcal{L},\mathcal{G}} := \{f : \mathbb{R}^d \to \mathbb{R}; \ \exists \mathbf{W} = (\mathbf{w}_1, ..., \mathbf{w}_k), \mathbf{w}_i \in L_{g(i)}, \ f(\mathbf{x}) = h(\mathbf{x}; \mathbf{W}), \ \forall \mathbf{x} \in \mathbb{R}^d\}. \quad (5)$$

We refer to this family of neural networks as Porcupine Neural Networks (PNNs). In general, functions described by PNNs (i.e., $\mathcal{F}_{\mathcal{L},\mathcal{G}}$) can be viewed as angular discretizations of functions described by unconstrained neural networks (i.e., $\mathcal{F}$). By increasing the size of $|\mathcal{L}|$ (i.e., the number of lines), we can approximate every $f \in \mathcal{F}$ by some $\hat{f} \in \mathcal{F}_{\mathcal{L},\mathcal{G}}$ arbitrarily closely. Thus, characterizing the landscape of the loss function over PNNs can help us to understand the landscape of the unconstrained loss function.

The PNN optimization can be written as

$$\min_{\mathbf{W}} \quad L(\mathbf{W}) \quad (6)$$
$$\mathbf{w}_i \in L_{g(i)} \quad 1 \leq i \leq k.$$

Matched and mismatched PNN optimizations are defined in a similar manner to the unconstrained ones.

## 4 Population Risk Landscapes of Matched PNNs

We say a vector has a positive orientation if its component in the largest non-zero index is positive. Otherwise, it has a negative orientation. For example, $\mathbf{w}_1 = (-1, 2, 0, 3, 0)$ has a positive orientation because $\mathbf{w}_1(4) > 0$, while the vector $\mathbf{w}_2 = (-1, 2, 0, 0, -3)$ has a negative orientation because $\mathbf{w}_2(5) < 0$. Mathematically, let $\mu(\mathbf{w}_i)$ be the largest index of the vector $\mathbf{w}_i$ with a non-zero entry, i.e., $\mu(\mathbf{w}_i) = \arg\max_j(\mathbf{w}_i(j) \neq 0)$. We define $s(\mathbf{w}_i) = 1$ if $\mu(\mathbf{w}_i) > 0$, otherwise $s(\mathbf{w}_i) = -1$.

Let $\mathbf{u}_i$ be a unit norm vector on the line $L_i$ such that $s(\mathbf{u}_i) = 1$. Let $\mathbf{U}_{\mathcal{L}} = (\mathbf{u}_1, ..., \mathbf{u}_r)$. Let $\mathbf{A}_{\mathcal{L}} \in \mathbb{R}^{r \times r}$ be a matrix whose $(i, j)$-component is the angle between lines $L_i$ and $L_j$, i.e., $\mathbf{A}_{\mathcal{L}}(i, j) = \theta_{\mathbf{u}_i, \mathbf{u}_j}$. Moreover, let $\mathbf{K}_{\mathcal{L}} = \mathbf{U}_{\mathcal{L}}^t \mathbf{U}_{\mathcal{L}} = \cos[\mathbf{A}_{\mathcal{L}}]$.

We define

$$q_r := \sum_{i \in \mathcal{G}_r} \|\mathbf{w}_i\|, \ \ q_r^* := \sum_{i \in \mathcal{G}_r} \|\mathbf{w}_i^*\|, \quad (7)$$

where $\|.\|$ is the second norm operator. Also, we define $\mathbf{q} := (q_1, ..., q_d)^t$ and $\mathbf{q}^* := (q_1^*, ..., q_d^*)^t$.

Define the kernel function $\psi : [-1, 1] \to \mathbb{R}$ as

$$\psi(x) = x + \frac{2}{\pi}\left(\sqrt{1 - x^2} - x\cos^{-1}(x)\right). \quad (8)$$

Some properties of this kernel function are explained in SM Section 4. In the following Theorem, we show that this kernel function plays an important role in characterizing optimizers of optimization (6). In particular, we show that the objective function of the neural network optimization has a term where this kernel function is applied (component-wise) to the inner product matrix among vectors $\mathbf{u}_1,...,\mathbf{u}_r$.

**Theorem 1** *The loss function* (3) *for a matched PNN can be written as*

$$L(\mathbf{W}) = \frac{1}{4}\|\sum_{i=1}^k \mathbf{w}_i - \mathbf{w}_i^*\|^2 + \frac{1}{4}(\mathbf{q} - \mathbf{q}^*)^t \psi[\mathbf{K}_{\mathcal{L}}](\mathbf{q} - \mathbf{q}^*), \quad (9)$$

*where $\psi(.)$ is defined as in* (8) *and $\mathbf{q}$ and $\mathbf{q}^*$ are defined as in* (7).

The kernel function $\psi(.)$ has a linear term and a nonlinear term. Note that the inner product matrix $\mathbf{K}_{\mathcal{L}}$ is positive semidefinite. Below, we show that applying the kernel function $\psi(.)$ (component-wise) to $\mathbf{K}_{\mathcal{L}}$ preserves this property.

**Lemma 1** *For every $\mathcal{L}$, $\psi[\mathbf{K}_{\mathcal{L}}]$ is positive semidefinite.*

Next, we characterize local optimizers of optimization (6) for a general PNN. Define $R(\mathbf{s}_1, ..., \mathbf{s}_r)$ as the space of $\mathbf{W}$ where $\mathbf{s}_i$ is the sign vector of weights $\mathbf{w}_j$ over the line $L_i$ (i.e., $j \in \mathcal{G}_i$).

**Theorem 2** *For a general PNN, in regions $R(\mathbf{s}_1, ..., \mathbf{s}_r)$ where at least $d$ of the $\mathbf{s}_i$'s are not equal to $\pm(1, 1, ..., 1)$, every local optimizer of optimization (6) is a global optimizer.*

In practice the number of lines $r$ is much larger than the number of inputs $d$ (i.e., $r \gg d$). Thus, the condition of Theorem 2 which requires $d$ out of $r$ variables $\mathbf{s}_i$ not to be equal to $\pm \mathbf{1}$ is likely to be satisfied if we initialize the local search algorithm randomly (SM Section 5).

# 5 Population Risk Landscapes of Mismatched PNNs

In this section, we characterize the population risk landscape of a mismatched PNN optimization where the model that generates the data and the model used in the PNN optimization are different. We assume that the output variable $y$ is generated using a two-layer PNN with $k^*$ neurons whose weights lie on the set of lines $\mathcal{L}^*$ with neuron-to-line mapping $\mathcal{G}^*$. That is

$$y = \sum_{i=1}^{k^*} \text{ReLU}\left( (\mathbf{w}_i^*)^t \mathbf{x} \right), \tag{10}$$

where $\mathbf{w}_i^*$ lies on a line in the set $\mathcal{L}^*$ for $1 \le i \le k^*$. The neural network optimization (6) is over PNNs with $k$ neurons over the set of lines $\mathcal{L}$ with the neuron-to-line mapping $\mathcal{G}$. Note that $\mathcal{L}$ and $\mathcal{G}$ can be different than $\mathcal{L}^*$ and $\mathcal{G}^*$, respectively.

Let $r = |\mathcal{L}|$ and $r^* = |\mathcal{L}^*|$ be the number of lines in $\mathcal{L}$ and $\mathcal{L}^*$, respectively. Let $\mathbf{u}_i^*$ be the unit norm vector on the line $L_i^* \in \mathcal{L}^*$ such that $s(\mathbf{u}_i^*) = 1$. Similarly, we define $\mathbf{u}_i$ as the unit norm vector on the line $L_i \in \mathcal{L}$ such that $s(\mathbf{u}_i) = 1$. Let $\mathbf{U}_{\mathcal{L}} = (\mathbf{u}_1, ..., \mathbf{u}_r)$ and $\mathbf{U}_{\mathcal{L}^*} = (\mathbf{u}_1^*, ..., \mathbf{u}_r^*)$. Suppose the rank of $\mathbf{U}_{\mathcal{L}}$ is at least $d$. Define

$$\mathbf{K}_{\mathcal{L}} = \mathbf{U}_{\mathcal{L}}^t \mathbf{U}_{\mathcal{L}} \in \mathbb{R}^{r \times r} \tag{11}$$

$$\mathbf{K}_{\mathcal{L}^*} = \mathbf{U}_{\mathcal{L}^*}^t \mathbf{U}_{\mathcal{L}^*} \in \mathbb{R}^{r^* \times r^*}$$

$$\mathbf{K}_{\mathcal{L},\mathcal{L}^*} = \mathbf{U}_{\mathcal{L}}^t \mathbf{U}_{\mathcal{L}^*} \in \mathbb{R}^{r \times r^*}.$$

**Theorem 3** *The loss function (3) for a mismatched PNN can be written as*

$$L(\mathbf{W}) = \frac{1}{4} \| \sum_{i=1}^{k} \mathbf{w}_i - \sum_{i=1}^{k^*} \mathbf{w}_i^* \|^2 + \frac{1}{4} \mathbf{q}^t \psi[\mathbf{K}_{\mathcal{L}}] \mathbf{q} + \frac{1}{4} (\mathbf{q}^*)^t \psi[\mathbf{K}_{\mathcal{L}^*}] \mathbf{q}^* - \frac{1}{2} \mathbf{q}^t \psi[\mathbf{K}_{\mathcal{L},\mathcal{L}^*}] \mathbf{q}^*, \tag{12}$$

*where $\psi(.)$ is defined as in (8) and $\mathbf{q}$ and $\mathbf{q}^*$ are defined as in (7) using $\mathcal{G}$ and $\mathcal{G}^*$, respectively.*

**Corollary 1** *Let*

$$\mathbf{K} = \begin{pmatrix} \mathbf{K}_{\mathcal{L}} & \mathbf{K}_{\mathcal{L},\mathcal{L}^*} \\ \mathbf{K}_{\mathcal{L},\mathcal{L}^*}^t & \mathbf{K}_{\mathcal{L}^*} \end{pmatrix} \in \mathbb{R}^{(r+r^*) \times (r+r^*)}. \tag{13}$$

*Then, the loss function of a mismatched PNN can be lower bounded as*

$$L(\mathbf{W}) \ge \frac{1}{4} \| \mathbf{q}^* \|^2 \lambda_{min} \left( \psi[\mathbf{K}] / \psi[\mathbf{K}_{\mathcal{L}}] \right) \tag{14}$$

*where $\psi[\mathbf{K}]/\psi[\mathbf{K}_{\mathcal{L}}] := \psi[\mathbf{K}_{\mathcal{L}^*}] - \psi[\mathbf{K}_{\mathcal{L}^*}]^t \psi[\mathbf{K}_{\mathcal{L}}]^\dagger \psi[\mathbf{K}_{\mathcal{L}^*}]$ is the generalized Schur complement of the block $\psi[\mathbf{K}_{\mathcal{L}}]$ in the matrix $\psi[\mathbf{K}]$.*

Next, we characterize local optimizers of optimization (6) for a mismatched PNN. Similar to the matched PNN case, we define $R(\mathbf{s}_1, ..., \mathbf{s}_r)$ as the space of $\mathbf{W}$ where $\mathbf{s}_i$ is the vector of sign variables of weight vectors over the line $L_i$.

**Theorem 4** *For a mismatched PNN, in regions $R(\mathbf{s}_1, ..., \mathbf{s}_r)$ where at least $d$ of the $\mathbf{s}_i$'s are not equal to $\pm(1, 1, ..., 1)$, every local optimizer of optimization* (6) *is a global optimizer. Moreover, in those points we have*

$$L(\mathbf{W}^*) = \frac{1}{4}(\mathbf{q}^*)^t \left(\psi[\mathbf{K}]/\psi[\mathbf{K}_{\mathcal{L}}]\right) \mathbf{q}^* \leq \frac{1}{4}\|\mathbf{q}^*\|^2 \|\psi[\mathbf{K}]/\psi[\mathbf{K}_{\mathcal{L}}]\|. \tag{15}$$

When the condition of Theorem 4 holds, the spectral norm of the matrix $\|\psi[\mathbf{K}]/\psi[\mathbf{K}_{\mathcal{L}}]\|$ provides an upper-bound on the loss value at global optimizers of the mismatched PNN. In Section 6, we study this bound in more detail. Moreover, in SM Section 7, we study the case where the condition of Theorem 4 does not hold (i.e., the local search method converges to a point in parameter regions where more than $r - d$ of variables $\mathbf{s}_i$ are equal to $\pm\mathbf{1}$). Finally, we study a Minimax analysis of the naive nearest line approximation approach in SM Section 8.

## 6 PNN Approximations of Unconstrained Neural Networks

In this section, we study whether an unconstrained two-layer neural network function can be approximated by a PNN. We assume that the unconstrained neural network has $d$ inputs and $k^*$ hidden neurons. This neural network function can also be viewed as a PNN whose lines are determined by input weight vectors to neurons. Thus, in this case $r^* \leq k^*$ where $r^*$ is the number of lines of the original network. If weights are generated randomly, with probability one, $r^* = k^*$ since the probability that two random vectors lie on the same line is zero. Note that lines of the ground-truth PNN (i.e., the unconstrained neural network) are unknowns in the training step. For training, we use a two-layer PNN with $r$ lines, drawn uniformly at random, and $k$ neurons. Since we have ReLU activation functions at neurons, without loss of generality, we can assume $k = 2r$, i.e., for every line we assign two neurons (one for potential weight vectors with positive orientations on that line and the other one for potential weight vectors with negative orientations). Since there is a mismatch between the model generating the data and the model used for training, we will have an approximation error. In this section, we study this approximation error as a function of parameters $d$, $r$ and $r^*$.

Since these lines will be different than $\mathcal{L}^*$, the neural network optimization can be formulated as a mismatched PNN optimization as studied in Section 5. Moreover, in this section, we assume the condition of Theorem 4 holds, i.e., the local search algorithm converges to a point in parameter regions where at least $d$ of variables $\mathbf{s}_i$ are not equal to $\pm\mathbf{1}$. The case that violates this condition is more complicated and is investigated in SM Section 7.

Under the condition of Theorem 4, the PNN approximation error depends on both $\|\mathbf{q}^*\|$ and $\|\psi[\mathbf{K}]/\psi[\mathbf{K}_{\mathcal{L}}]\|$. The former term provides a scaling normalization for the loss function. Thus, we focus on analyzing the later term.

Since Theorem 4 provides an upper-bound for the mismatched PNN optimization loss using $\|\psi[\mathbf{K}]/\psi[\mathbf{K}_{\mathcal{L}}]\|$, intuitively, increasing the number of lines in $\mathcal{L}$ should decrease $\|\psi[\mathbf{K}]/\psi[\mathbf{K}_{\mathcal{L}}]\|$. We prove this in the following theorem.

**Theorem 5** *Let $\mathbf{K}$ be defined as in* (13). *We add a distinct line to the set $\mathcal{L}$, i.e., $\mathcal{L}_{new} = \mathcal{L} \cup L_{r+1}$. Define*

$$\mathbf{K}_{new} = \begin{pmatrix} \mathbf{K}_{\mathcal{L}_{new}} & \mathbf{K}_{\mathcal{L}_{new}, \mathcal{L}^*} \\ \mathbf{K}_{\mathcal{L}_{new}, \mathcal{L}^*}^t & \mathbf{K}_{\mathcal{L}^*} \end{pmatrix} = \begin{pmatrix} 1 & \mathbf{z}_1^t & \mathbf{z}_2^t \\ \mathbf{z}_1 & \mathbf{K}_{\mathcal{L}} & \mathbf{K}_{\mathcal{L}, \mathcal{L}^*} \\ \mathbf{z}_2 & \mathbf{K}_{\mathcal{L}, \mathcal{L}^*}^t & \mathbf{K}_{\mathcal{L}^*} \end{pmatrix} \in \mathbb{R}^{(r+r^*+1) \times (r+r^*+1)}. \tag{16}$$

*Then, we have*

$$\|\psi[\mathbf{K}_{new}]/\psi[\mathbf{K}_{\mathcal{L}_{new}}]\| \leq \|\psi[\mathbf{K}]/\psi[\mathbf{K}_{\mathcal{L}}]\|. \tag{17}$$

*More specifically,*

$$\psi[\mathbf{K}_{new}]/\psi[\mathbf{K}_{\mathcal{L}_{new}}] = \psi[\mathbf{K}]/\psi[\mathbf{K}_{\mathcal{L}}] - \alpha\mathbf{v}\mathbf{v}^t, \tag{18}$$

*where $\alpha = \left(1 - \langle\psi[\mathbf{z}_1], \psi[\mathbf{K}_{\mathcal{L}}]^{-1}\psi[\mathbf{z}_1]\rangle\right)^{-1} \geq 0$, $\mathbf{v} = \psi[\mathbf{z}_2] - \psi[\mathbf{K}_{\mathcal{L}, \mathcal{L}^*}]^t\psi[\mathbf{K}_{\mathcal{L}}]^{-1}\psi[\mathbf{z}_1]$.*

Theorem 5 indicates that adding lines to $\mathcal{L}$ decreases $\|\psi[\mathbf{K}]/\psi[\mathbf{K}_{\mathcal{L}}]\|$. However, it does not characterize the rate of this decrease as a function of $r$, $r^*$ and $d$.

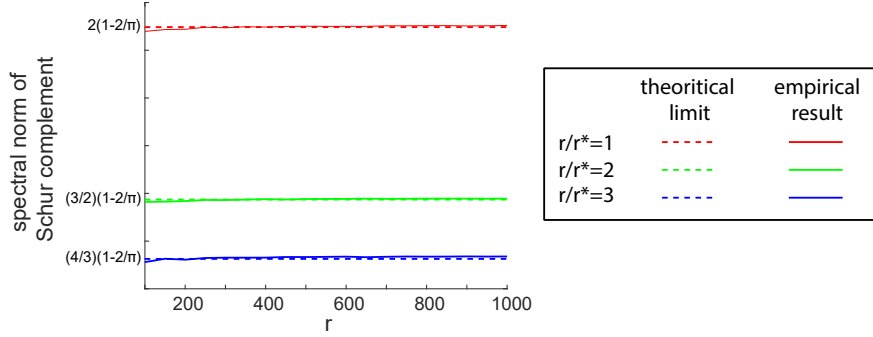

Figure 2: The spectral norm of $\psi[\mathbf{K}]/\psi[\mathbf{K}_{\mathcal{L}}]$ when $d = r$. Theoretical limits are described in Theorem 6. Experiments have been repeated 100 times. Average results are shown.

Next, we characterize the asymptotic behaviour of $\|\psi[\mathbf{K}]/\psi[\mathbf{K}_{\mathcal{L}}]\|$ when $d, r \to \infty$. There has been some recent interest in characterizing spectrum of inner product kernel random matrices [31, 32, 33, 34]. If the kernel is linear, the distribution of eigenvalues of the covariance matrix follows the well-known Marcenko-Pastur law. If the kernel is nonlinear, reference [31] shows that in the high dimensional regime where $d, r \to \infty$ and $\gamma = r/d \in (0, \infty)$ is fixed, only the linear part of the kernel function affects the spectrum. Note that the matrix of interest in our problem is the Schur complement matrix $\psi[\mathbf{K}]/\psi[\mathbf{K}_{\mathcal{L}}]$, not $\psi[\mathbf{K}]$. However, we can use results characterizing the spectrum of $\psi[\mathbf{K}]$ to characterize the spectrum of $\psi[\mathbf{K}]/\psi[\mathbf{K}_{\mathcal{L}}]$.

We consider the regime where $r, d \to \infty$ while $\gamma = r/d \in (0, \infty)$ is a fixed number. Theorem 2.1 of reference [31] shows that in this regime and under some mild assumptions on the kernel function (which our kernel function $\psi(.)$ satisfies), $\psi[\mathbf{K}_{\mathcal{L}}]$ converges (in probability) to the following matrix:

$$\mathbf{R}_{\mathcal{L}} = \left( \psi(0) + \frac{\psi''(0)}{2d} \right) \mathbf{1}\mathbf{1}^t + \psi'(0)\mathbf{U}_{\mathcal{L}}^t\mathbf{U}_{\mathcal{L}} + (\psi(1) - \psi(0) - \psi'(0))\,\mathbf{I}_r. \tag{19}$$

To obtain this formula, one can write the Taylor expansion of the kernel function $\psi(.)$ near 0. It turns out that in the regime where $r, d \to \infty$ while $d/r$ is fixed, it is sufficient for off-diagonal elements of $\psi[\mathbf{K}_{\mathcal{L}}]$ to replace $\psi(.)$ with its linear part. However, diagonal elements of $\psi[\mathbf{K}_{\mathcal{L}}]$ should be adjusted accordingly (the last term in (19)). For the kernel function of our interest, defined as in (8), we have $\psi'(0) = 0$, $\psi''(0) = 2/\pi$, $\psi(0) = 2/\pi$ and $\psi(1) = 1$. This simplifies (19) further to:

$$\mathbf{R}_{\mathcal{L}} = \left( \frac{2}{\pi} + \frac{1}{\pi d} \right) \mathbf{1}\mathbf{1}^t + (1 - \frac{2}{\pi})\mathbf{I}_r. \tag{20}$$

This matrix has $(r - 1)$ eigenvalues of $1 - 2/\pi$ and one eigenvalue of $(2/\pi)r + 1 - 2/\pi + \gamma/\pi$. Using this result, we characterize $\|\psi[\mathbf{K}]/\psi[\mathbf{K}_{\mathcal{L}}]\|$ in the following theorem:

**Theorem 6** *Let $\mathcal{L}$ and $\mathcal{L}^*$ have $r$ and $r^*$ lines in $\mathbb{R}^d$ generated uniformly at random, respectively. Let $d, r \to \infty$ while $\gamma = r/d \in (0, \infty)$ is fixed. Moreover, $r^*/r = \mathcal{O}(1)$. Then,*

$$\|\psi[\mathbf{K}]/\psi[\mathbf{K}_{\mathcal{L}}]\| \to \left( 1 + \frac{r^*}{r} \right) \left( 1 - \frac{2}{\pi} \right), \tag{21}$$

*where the convergence is in probability.*

Figure 2 shows the spectral norm of $\psi[\mathbf{K}]/\psi[\mathbf{K}_{\mathcal{L}}]$ when $d = r$. As it is illustrated in this figure, empirical results match closely to analytical limits of Theorem 6. Note that by increasing the ratio of $r/r^*$, $\|\psi[\mathbf{K}]/\psi[\mathbf{K}_{\mathcal{L}}]\|$ and therefore the PNN approximation error decreases. If $r^*$ is constant, the limit is $1 - 2/\pi \approx 0.36$.

Theorem 6 provides a bound on $\|\psi[\mathbf{K}]/\psi[\mathbf{K}_{\mathcal{L}}]\|$ in the asymptotic regime. In the following corollary, we use this result to bound the PNN approximation error measured as the MSE normalized by the $L_2$ norm of the output variables (i.e., $L(\mathbf{W} = 0)$).

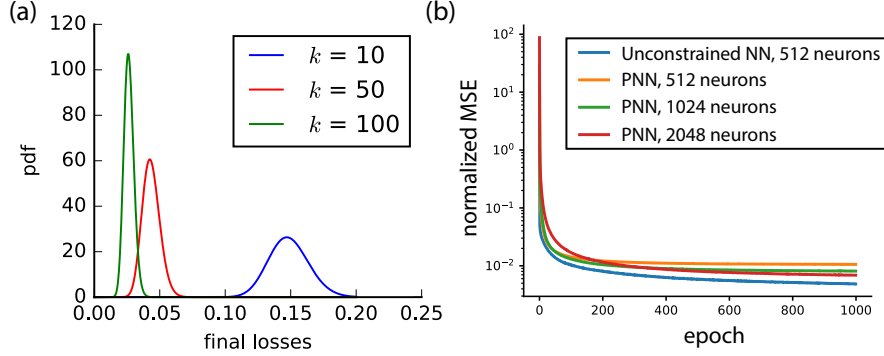

Figure 3: (a) Gamma curves fit to Histogram of PNN approximation errors for different values of $k$. (b) Normalized losses obtained by training unconstrained and PNNs on the MNIST dataset.

**Proposition 1** *Let* $\mathbf{W}^*$ *be the global optimizer of the mismatched PNN optimization under the setup of Theorem 6. Then, with high probability, we have*

$$\frac{L(\mathbf{W}^*)}{L(\mathbf{W} = \mathbf{0})} \leq \left(1 + \frac{r^*}{r}\right)\left(1 - \frac{2}{\pi}\right). \tag{22}$$

This proposition indicates that in the asymptotic regime with $d$ and $r$ growing with the same rate (i.e., $r$ is a constant factor of $d$), the PNN is able to explain a fraction of the variance of the output variable. In practice, however, $r$ should grow faster than $d$ in order to obtain a small PNN approximation error.

## 7 Experiments on Synthetic and MNIST Datasets

In this section, we numerically evaluate the performance of PNNs on synthetic and MNIST datasets. We use random PNNs as described in Section 5. To enforce the PNN architecture, we project gradients along the directions of the PNN lines before updating the weights. For more details on these experiments, see SM Section 9.

In the synthetic data experiments, we generated data using a fully-connected two-layer network with $d = 15$ inputs and $k^* = 20$ hidden neurons. We generated 10,000 ground-truth training samples and 10,000 test samples using a set of randomly chosen weights for the network. For PNNs, we use $10 \leq k \leq 100$ hidden neurons. For each value of $k$, we perform 25 trials. We train the PNN via stochastic gradient descent using batches of size 100, 100 training epochs, no momentum, and a learning rate of $10^{-3}$ which decays every epoch at a rate of 0.95 every 390 epochs. For evaluation, we compute the normalized MSE (i.e., MSE normalized by the $L_2$ norm of $y$) in the test set over different initializations. The results are shown in Figures 1-b and 3. Figure 3-a shows that as $k$ increases, the PNN approximation gets better, which is consistent to our theoretical results in Section 6.

Next, we evaluate PNNs on MNSIT. We first trained a dense network on a subset of the MNIST handwritten digits dataset. Of the 10 types of 28x28 MNIST images, we only looked at images of 1's and 2's, assigning them the labels of $y = 1$ and $y = 2$, respectively. This resulted in $n = 11,649$ training samples and $2,167$ test samples. The network has the structure shown in Figure 1-a except weight vectors in the first layer are not constrained. Only first-layer weights were updated during training. The unconstrained network has $k^* = 512$ hidden neurons.

We then approximated this trained network using PNNs of similar structures as previously described. We tested $k = 512, 1024, 2048$ (recall that each $k$ involves $r = k/2$ unique lines). Figure 3-b shows the normalized losses obtained using unconstrained networks and PNNs. As illustrated in this figure, the PNN approximation error is small even with similar numbers of neurons to that of the unconstrained network.

# 8 Conclusion

In this paper, we introduced a family of constrained neural networks, called Porcupine Neural Networks (PNNs), whose population risk landscapes have good theoretical properties, i.e., most local optima of PNN optimizations are global while we have a characterization of parameter regions where bad local optimizers may exist. We also showed that an unconstrained (fully-connected) neural network function can be approximated by a polynomially-large PNN. In particular, we provided approximation bounds at global optima and also bad local optima (under some conditions) of the PNN optimization. These results may provide a means of explaining the success of local search methods in solving the unconstrained neural network optimization because every bad local optimum of the unconstrained problem can be viewed as a local optimum (either good or bad) for a PNN constrained problem, which has a bounded loss according to our results. We leave further explorations of this idea for future work.

In Section 6, we used a set of projection lines for PNNs that are generated uniformly randomly. The choice of the projection lines will affect the kernel matrix and thus can affect the capacity of a PNN. A more sophisticated design can have a higher density of projection lines along directions with high variances. Exploring the design of projection lines for PNNs can be an interesting direction for future work. In addition, it is possible to extend our results to feed-forward neural networks with more than two layers by constraining weight vectors in every layer, except the last one, to lie on fixed projection lines. In that case and using the method proposed in [26], we can generalize our analysis by replacing the kernel function of (8) with a kernel that also depends on the architecture. Finally, other extensions of PNNs to network architectures with different activation functions than ReLU, and other types of operations (such as convolutions) are also among interesting directions for future work.

# 9 Code

We provide code for PNN experiments in the following link:

`https://github.com/jessemzhang/porcupine_neural_networks`

# 10 Acknowledgment

This work supported by the Center for Science of Information (CSoI), an NSF Science and Technology Center, under grant agreement CCF-0939370.

## Footnotes

[1]In this document, we refer to pointers in the supplementary materials using the prefix *SM*.

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
