[Supplementary Material · SM.pdf]

Supplementary Materials
# Porcupine Neural Networks:
# Approximating Neural Network Landscapes

**Soheil Feizi**
Department of Computer Science
University of Maryland, College Park
sfeizi@cs.umd.edu

**Hamid Javadi**
Department of Electrical and Computer Engineering
Rice University
hrhakim@rice.edu

**Jesse Zhang**
Department of Electrical Engineering
Stanford University
jessez@stanford.edu

**David Tse**
Department of Electrical Engineering
Stanford University
dntse@stanford.edu

## Contents

# 1 Notation

In this document, we refer to pointers in the main text using the prefix *MT*. For example, equation MT-1 refers to equation 1 in the main text.

For matrices we use bold-faced upper case letters, for vectors we use bold-faced lower case letters, and for scalars we use regular lower case letters. For example, $\mathbf{X}$ represents a matrix, $\mathbf{x}$ represents a vector, and $x$ represents a scalar number. $\mathbf{I}_n$ is the identity matrix of size $n \times n$. $\mathbf{e}_j$ is a vector whose $j$-th element is non-zero and its other elements are zero. $\mathbf{1}_{n_1,n_2}$ is the all one matrix of size $n_1 \times n_2$. When no confusion arises, we drop the subscripts. $\mathbf{1}\{x = y\}$ is the indicator function which is equal to one if $x = y$, otherwise it is zero. $\mathrm{ReLU}(x) = \max(x, 0)$. $Tr(\mathbf{X})$ and $\mathbf{X}^t$ represent the trace and the transpose of the matrix $\mathbf{X}$, respectively. $\|\mathbf{x}\|_2 = \mathbf{x}^t \mathbf{x}$ is the second norm of the vector $\mathbf{x}$. When no confusion arises, we drop the subscript. $\|\mathbf{x}\|_1$ is the $l_1$ norm of the vector $\mathbf{x}$. $\|\mathbf{X}\|$ is the operator (spectral) norm of the matrix $\mathbf{X}$. $\|\mathbf{x}\|_0$ is the number of non-zero elements of the vector $\mathbf{x}$. $< \mathbf{x}, \mathbf{y} >$ is the inner product between vectors $\mathbf{x}$ and $\mathbf{y}$. $\mathbf{x} \perp \mathbf{y}$ indicates that vectors $\mathbf{x}$ and $\mathbf{y}$ are orthogonal. $\theta_{\mathbf{x},\mathbf{y}}$ is the angle between vectors $\mathbf{x}$ and $\mathbf{y}$. $\mathcal{N}(\mu, \mathbf{\Gamma})$ is the Gaussian distribution with mean $\mu$ and the covariance $\mathbf{\Gamma}$. $f[\mathbf{A}]$ is a matrix where the function $f(.)$ is applied to its components, i.e., $f[\mathbf{A}](i,j) = f(\mathbf{A}(i,j))$. $\mathbf{A}^\dagger$ is the pseudo inverse of the matrix $\mathbf{A}$. The eigen decomposition of the matrix $\mathbf{A} \in \mathbb{R}^{n \times n}$ is denoted by $\mathbf{A} = \sum_{i=1}^{n} \lambda_i(\mathbf{A})\mathbf{u}_i(\mathbf{A})\mathbf{u}_i(\mathbf{A})^t$, where $\lambda_i(\mathbf{A})$ is the $i$-th largest eigenvalue of the matrix $\mathbf{A}$ corresponding to the eigenvector $\mathbf{u}_i(\mathbf{A})$. We have $\lambda_1(\mathbf{A}) \geq \lambda_2(\mathbf{A}) \geq \cdots$.

# 2 Related Work

To explain the success of neural networks, some references study their ability to approximate smooth functions [1, 2, 3, 4, 5, 6, 7], while some other references focus on benefits of having more layers [8, 9]. Over-parameterized networks where the number of parameters are larger than the number of training samples have been studied in [10, 11]. However, such architectures can cause generalization issues in practice [12].

References [13, 14, 15, 16] have studied the convergence of the local search algorithms such as gradient descent methods to the global optimum of the neural network optimization with zero hidden neurons and a single output. In this case, the loss function of the neural network optimization has a

Figure 1: Examples of (a) scalar PNN, and (b) degree-one PNN structures.

single local optimizer which is the same as the global optimum. However, for neural networks with hidden neurons, the landscape of the loss function is more complicated than the case with no hidden neurons.

Several work has studied the risk landscape of neural network optimizations for more complex structures under various model assumptions [17, 18, 19, 20, 21, 22, 23, 24, 25, 26, 27]. Reference [17] shows that in the linear neural network optimization, the population risk landscape does not have any bad local optima. Reference [18] extends these results and provides necessary and sufficient conditions for a critical point of the loss function to be a global minimum. Reference [19] shows that for a two-layer neural network with leaky activation functions, the gradient descent method on a modified loss function converges to a global optimizer of the modified loss function which can be different from the original global optimum. Under an independent activations assumption, reference [20] simplifies the loss function of a neural network optimization to a polynomial and shows that local optimizers obtain approximately the same objective values as the global ones. This result has been extended by reference [17] to show that all local minima are global minima in a nonlinear network. However, the underlying assumption of having independent activations at neurons usually are not satisfied in practice.

References [21, 22, 23] consider a two-layer neural network with Gaussian inputs under a matched (realizable) model where the output is generated from a network with planted weights. Moreover, they assume the number of neurons in the hidden layer is smaller than the dimension of inputs. This critical assumption makes the loss function positive-definite in a small neighborhood near the global optimum. Then, reference [23] provides a tensor-based method to initialize the local search algorithm in that neighborhood which guarantees its convergence to the global optimum. In our problem formulation, the number of hidden neurons can be larger than the dimension of inputs as it is often the case in practice. Moreover, we characterize risk landscapes for a certain family of neural networks in all parameter regions, not just around the global optimizer. This can guide us towards understanding the reason behind the success of local search methods in practice.

For a neural network with a single non-overlapping convolutional layer, reference [24] shows that all local optimizers of the loss function are global optimizers as well. They also show that in the overlapping case, the problem is NP-hard when inputs are not Gaussian. Moreover, reference [25] studies this problem with non-standard activation functions, while reference [26] considers the case where the weights from the hidden layer to the output are close to the identity. Other related works include improper learning models using kernel based approaches [28, 29] and a method of moments estimator using tensor decomposition [27].

## 3 PNN Examples Imposed by the Network Architecture

In some cases, the PNN constraint is imposed by the neural network architecture. For example, consider the neural network depicted in Figure 1-a, which has a single input and $k$ neurons. In this network structure, $\mathbf{w}_i$'s are scalars. Thus, every realizable function with this neural network can be realized using a PNN where $\mathcal{L}$ includes a single line. We refer to this family of neural networks as scalar PNNs. Another example of porcupine neural networks is depicted in Figure 1-b. In this case, the neural network has multiple inputs and multiple neurons. Each neuron in this network is

connected to one input. Every realizable function with this neural network can be described using a PNN whose lines are parallel to standard axes. We refer to this family of neural networks as degree-one PNNs. Scalar PNNs are also degree-one PNNs. However, since their analysis is simpler, we make such a distinction.

Below we characterize landscape properties of scalar and degree-one PNNs in the matched case.

### 3.1 Scalar PNNs

In this section, we consider a neural network structure with a single input and multiple neurons (i.e., $d = 1$, $k > 1$). Such neural networks are PNNs with $\mathcal{L}$ containing a single line. Thus, we refer to them as scalar PNNs. An example of a scalar PNN is depicted in Figure 1-a. In this case, every $\mathbf{w}_i$ for $1 \leq i \leq k$ is a single scalar. We refer to that element by $w_i$. We assume $w_i$'s are non-zero, otherwise the neural network structure can be reduced to another structure with fewer neurons.

**Theorem 1** *The loss function MT-(3) for a scalar PNN can be written as*

$$L(\mathbf{W}) = \frac{1}{4}\left(\sum_{i=1}^{k} w_i - \sum_{i=1}^{k} w_i^*\right)^2 + \frac{1}{4}\left(\sum_{i=1}^{k} |w_i| - \sum_{i=1}^{k} |w_i^*|\right)^2. \tag{1}$$

Since for a scalar PNN, the loss function $L(\mathbf{W})$ can be written as sum of squared terms, we have the following corollary:

**Corollary 1** *For a scalar PNN, $\mathbf{W}$ is the global optimizer of optimization MT-(6) if and only if*

$$\sum_{i=1}^{k} w_i = \sum_{i=1}^{k} w_i^*, \tag{2}$$

$$\sum_{i=1}^{k} |w_i| = \sum_{i=1}^{k} |w_i^*|.$$

Next, we characterize local optimizers of optimization MT-(6).

Let $s(w_i)$ be the sign variable of $w_i$, i.e., $s(w_i) = 1$ if $w_i > 0$, otherwise $s(w_i) = -1$. Let $s(\mathbf{W}) \triangleq (s(w_1), ..., s(w_k))^t$. Let $R(\mathbf{s})$ denote the space of all $\mathbf{W}$ where $s_i = s(w_i)$, i.e., $R(\mathbf{s}) \triangleq \{(w_1, ..., w_k) : s(w_i) = s_i\}$.

**Theorem 2** *If $s(\mathbf{W}^*) \neq \pm\mathbf{1}$:*

- *In every region $R(\mathbf{s})$ whose $\mathbf{s} \neq \pm\mathbf{1}$, optimization MT-(6) only has global optimizers without any bad local optimizers.*

- *In two regions $R(\mathbf{1})$ and $R(-\mathbf{1})$, optimization MT-(6) does not have global optimizers and only has bad local optimizers.*

*If $s(\mathbf{W}^*) = \pm\mathbf{1}$:*

- *In regions $R(\mathbf{s})$ where $\mathbf{s} \neq \pm\mathbf{1}$ and in the region $R(-s(\mathbf{W}^*))$, optimization MT-(6) neither has global nor bad local optimizers.*

- *In the region $R(s(\mathbf{W}^*))$, optimization MT-(6) only has global optimizers without any bad local optimizers.*

Theorem 2 indicates that optimization MT-(6) can have bad local optimizers. However, this can occur only in two parameter regions, out of $2^k$ regions, which can be checked separately (Figure 2). Thus, a variant of the gradient descent method which checks these cases separately converges to a global optimizer.

Next, we characterize the Hessian of the loss function:

Figure 2: For the scalar PNN, parameter regions where $s(\mathbf{W}) = \pm\mathbf{1}$ may include bad local optima. In other regions, all local optima are global. This figure highlights regions where $s(\mathbf{W}) = \pm\mathbf{1}$ for a scalar PNN with two neurons.

Figure 3: The landscape of the loss function for a scalar PNN with two neurons. In panel (a), we consider $w_1^* = 6$ and $w_2^* = 4$, while in panel (b), we have $w_1^* = 6$ and $w_2^* = -4$. According to Theorem 2, in the case of panel (a), the loss function does not have bad local optimizers, while in the case of panel (b), it has bad local optimizers in regions $R\left((-1,-1)\right)$ and $R\left((1,1)\right)$.

**Theorem 3** *For a scalar PNN, in every region $R(\mathbf{s})$, the Hessian matrix of the loss function $L(\mathbf{W})$ is positive semidefinite, i.e., in every region $R(\mathbf{s})$, the loss function is convex. In regions $R(\mathbf{s})$ where $\mathbf{s} \neq \pm\mathbf{1}$, the rank of the Hessian matrix is two, while in two regions $R(\pm\mathbf{1})$, the rank of the Hessian matrix is equal to one.*

Finally, for a scalar PNN, we illustrate the landscape of the loss function with an example. Figure 3 considers the case with a single input and two neurons (i.e., $d = 1$, $k = 2$). In Figure 3-a, we assume $w_1^* = 6$ and $w_2^* = 4$. According to Theorem 2, only the region $R\left((1,1)\right)$ contains global optimizers (all points in this region on the line $w_1 + w_2 = 10$ are global optimizers.). In Figure 3-b, we consider $w_1^* = 6$ and $w_2^* = -4$. According to Theorem 2, regions $R\left((1,-1)\right)$ and $R\left((-1,1)\right)$ have global optimizers, while regions $R\left((1,1)\right)$ and $R\left((-1,-1)\right)$ include bad local optimizers.

## 3.2  Degree-One PNNs

In this section, we consider a neural network structure with more than one input and multiple neurons ($d \geq 1$ and $k > 1$) such that each neuron is connected to one input. Such neural networks are PNNs whose lines are parallel to standard axes. Thus, we refer to them as degree-one PNNs.

Similar to the scalar PNN case, in the case of the degree-one PNN, every $\mathbf{w}_i$ has one non-zero element. We refer to that element by $w_i$. Let $\mathcal{G}_r$ be the set of neurons that are connected to the variable $x_r$, i.e., $\mathcal{G}_r = \{j : \mathbf{w}_j(r) \neq 0\}$. Therefore, we have $\mathcal{G}_1 \cup ... \cup \mathcal{G}_d = \{1,...,k\}$. Moreover,

Figure 4: (a) An example of $\mathcal{L}$ in a two-dimensional space such that angles between adjacent lines are equal to one another. (b) The minimum eigenvalue of the matrix $\psi[\mathbf{K}_{\mathcal{L}}]$ for different values of $r$.

we assume $\mathcal{G}_i \neq \emptyset$ for $1 \leq i \leq d$, i.e., there is at least one neuron connected to each input variable. For every $j \in \mathcal{G}_r$, we define the function $g(.)$ such that $g(j) = r$ [1].

**Theorem 4** *The loss function MT-(3) for a degree-one PNN can be written as*

$$L(\mathbf{W}) = \frac{1}{4}\|\sum_{i=1}^{k}\mathbf{w}_i - \sum_{i=1}^{k}\mathbf{w}_i^*\|^2 + \frac{1}{4}(\mathbf{q} - \mathbf{q}^*)^t\mathbf{C}(\mathbf{q} - \mathbf{q}^*), \tag{3}$$

*where*

$$\mathbf{C} = \begin{pmatrix} 1 & \frac{2}{\pi} & \cdots & \frac{2}{\pi} \\ \frac{2}{\pi} & 1 & \cdots & \frac{2}{\pi} \\ \vdots & & \ddots & \vdots \\ \frac{2}{\pi} & & \cdots & 1 \end{pmatrix}. \tag{4}$$

Since $\mathbf{C}$ is a positive definite matrix, we have the following corollary:

**Corollary 2** $\mathbf{W}^*$ *is a global optimizer of optimization MT-(6) for a degree-one PNN if and only if*

$$\sum_{i \in \mathcal{G}_r} w_i = \sum_{i \in \mathcal{G}_r} w_i^*, \quad 1 \leq r \leq d \tag{5}$$

$$q_i = q_i^*, \quad 1 \leq r \leq d.$$

Next, we characterize local optimizers of optimization MT-(6) for degree-one PNNs. The sign variable assigned to the weight vector $\mathbf{w}_j$ is defined as the sign of its non-zero element, i.e., $s(\mathbf{w}_j) = s(w_j)$ where $w_j$ is the non-zero element of $\mathbf{w}_j$. Define $R(\mathbf{s}_1, ..., \mathbf{s}_d)$ as the space of $\mathbf{W}$ where $\mathbf{s}_i$ is the sign vector of weights $\mathbf{w}_j$ connected to input $x_i$ (i.e., $j \in \mathcal{G}_i$).

**Theorem 5** *For a degree-one PNN, in regions $R(\mathbf{s}_1, ..., \mathbf{s}_d)$ where $\mathbf{s}_i \neq \pm\mathbf{1}$ for $1 \leq i \leq d$, every local optimizer is a global optimizer. In other regions, we may have bad local optima.*

In practice, if the gradient descent algorithm converges to a point in a region $R(\mathbf{s}_1, ..., \mathbf{s}_d)$ where signs of weight vectors connected to an input are all ones or minus ones, that point may be a bad local optimizer. Thus, one may re-initialize the gradient descent algorithm in such cases. We show this effect through simulations in Section 7.

# 4 Properties of the Kernel Function $\psi(.)$

**Example 1** Let $\mathcal{L} = \{L_1, L_2, ..., L_r\}$ contain lines in $\mathbb{R}^2$ such that angles between adjacent lines are equal to $\pi/r$ (Figure 4-a). Thus, we have $\mathbf{A}_{\mathcal{L}}(i, j) = \pi|i - j|/r$ for $1 \leq i, j \leq r$. Figure 4-b

shows the minimum eigenvalue of the matrix $\psi[\mathbf{K}_{\mathcal{L}}]$ for different values of $r$. As the number of lines increases, the minimum eigenvalue of $\psi[\mathbf{K}_{\mathcal{L}}]$ decreases. However, for a finite value of $r$, the minimum eigenvalue of $\psi[\mathbf{K}_{\mathcal{L}}]$ is positive. This highlights why considering a discretized neural network function (i.e., finite $r$) facilities characterizing the landscape of the loss function.

# 5    Number of Bad Parameter Regions of PNNs

Consider a two-layer PNN with $d$ inputs, $r$ lines and $k$ hidden neurons. Suppose every line corresponds to $t = k/r$ input weight vectors. If we generate weight vectors uniformly at random over their corresponding lines, for every $1 \le i \le r$, we have

$$\mathbb{P}[\mathbf{s}_i = \pm \mathbf{1}] = 2^{1-t}. \tag{6}$$

As $t$ increases, this probability decreases exponentially. According to Theorem MT-2, to be in the parameter region without bad locals, the event $\mathbf{s}_i = \pm \mathbf{1}$ should occur for at most $r - d$ of the lines. Thus, if we uniformly pick a parameter region, the probability of selecting a region without bad locals is

$$1 - \sum_{i=1}^{d-1} \binom{r}{i} (1 - 2^{1-t})^i 2^{(1-t)(r-i)} \tag{7}$$

which goes to one exponentially as $r \to \infty$.

In practice the number of lines $r$ is much larger than the number of inputs $d$ (i.e., $r \gg d$). Thus, the condition of Theorem MT-2 which requires $d$ out of $r$ variables $\mathbf{s}_i$ not to be equal to $\pm \mathbf{1}$ is likely to be satisfied if we initialize the local search algorithm randomly.

# 6    PNN Perturbation Analysis

In this section, we show that if $\mathbf{U}_{\mathcal{L}}$ is a perturbed version of $\mathbf{U}_{\mathcal{L}^*}$, the loss in global optima of the mismatched PNN optimization MT-(6) is small. This shows a continuity property of the PNN optimization with respect to line perturbations.

**Lemma 1** *Let $\mathbf{K}$ is defined as in MT-(13) where $r = r^*$. Let $\mathbf{Z} := \mathbf{U} - \mathbf{U}^*$ be the perturbation matrix. Assume that $\lambda_{\min}\left(\psi\left[\mathbf{K}_{\mathcal{L}^*}\right]\right) \ge \delta$. If*

$$2\sqrt{r}\|\mathbf{Z}\|_F + \|\mathbf{Z}\|_F^2 \le \frac{\delta}{2},$$

*then*

$$\|\psi[\mathbf{K}]/\psi[\mathbf{K}_{\mathcal{L}}]\|_2 \le \left(1 + \frac{2r}{\delta}\right)\|\mathbf{Z}\|_F^2 + 4\sqrt{r}\|\mathbf{Z}\|_F.$$

# 7    The General PNN Approximation Error

In this section, we consider the case where the condition of Theorem MT-4 does not hold, i.e., the local search algorithm converges to a point in a *bad* parameter region where more than $r - d$ of $\mathbf{s}_i$ variables are equal to $\pm \mathbf{1}$. To simplify notation, we assume that the local search method has converged to a region where all $\mathbf{s}_i$ variables are equal to $\pm \mathbf{1}$. The analysis extends naturally to other cases as well.

Let $\mathbf{s} = (\mathbf{s}_1, ..., \mathbf{s}_r)$. Let $\mathbf{S}$ be the diagonal matrix whose diagonal entries are equal to $\mathbf{s}$, i.e., $\mathbf{S} = \mathrm{diag}(\mathbf{s})$. Similar to the argument of Theorems MT-2 and MT-4, a necessary condition for a point $\mathbf{W}$ to be a local optima of the PNN optimization is:

$$\mathbf{S}\mathbf{U}_{\mathcal{L}}^t \left( \sum_{i=1}^{k} \mathbf{w}_i - \sum_{i=1}^{k^*} \mathbf{w}_i^* \right) + \psi[\mathbf{K}_{\mathcal{L}}]\mathbf{q} - \psi[\mathbf{K}_{\mathcal{L},\mathcal{L}^*}]\mathbf{q}^* = 0. \tag{8}$$

Under the condition of Theorem MT-4, we have $\sum_{i=1}^{k} \mathbf{w}_i - \sum_{i=1}^{k^*} \mathbf{w}_i^* = \mathbf{0}$, which simplifies this condition.

Using (8) in MT-(12), at local optima in bad parameter regions, we have

$$4L(\mathbf{W}) = (\mathbf{q}^*)^t \, \psi[\mathbf{K}]/\psi[\mathbf{K}_{\mathcal{L}}]\mathbf{q}^* + \mathbf{z}^t \left( \mathbf{I} + \mathbf{U}_{\mathcal{L}}\mathbf{S}\psi[\mathbf{K}_{\mathcal{L}}]^{-1}\mathbf{S}\mathbf{U}_{\mathcal{L}}^t \right) \mathbf{z}, \tag{9}$$

where

$$\mathbf{z} := \sum_{i=1}^{k} \mathbf{w}_i - \sum_{i=1}^{k^*} \mathbf{w}_i^*. \tag{10}$$

The first term of (9) is similar to the PNN loss under the condition of Theorem MT-4. The second term is the price paid for converging to a point in a bad parameter region. In this section, we analyze this term.

The second term of (9) depends on the norm of $\mathbf{z}$. First, in the following lemma, we characterize $\mathbf{z}$ in local optima.

**Lemma 2** *In the local optimum of the mismatched PNN optimization, we have*

$$\mathbf{z} = - \left( \mathbf{U}_{\mathcal{L}}\mathbf{S}\mathbf{S}^t\mathbf{U}_{\mathcal{L}}^t \right)^{-1} \mathbf{U}_{\mathcal{L}}\mathbf{S}\bigg[ \psi[\mathbf{K}_{\mathcal{L}}] \left( \mathbf{S}\mathbf{U}_{\mathcal{L}}^t\mathbf{U}_{\mathcal{L}}\mathbf{S} + \psi[\mathbf{K}_{\mathcal{L}}] \right)^\dagger \mathbf{S}\mathbf{U}_{\mathcal{L}}^t\mathbf{w}_0 \tag{11}$$

$$+ \left( \psi[\mathbf{K}_{\mathcal{L}}] \left( \mathbf{S}\mathbf{U}_{\mathcal{L}}^t\mathbf{U}_{\mathcal{L}}\mathbf{S} + \psi[\mathbf{K}_{\mathcal{L}}] \right)^\dagger - \mathbf{I} \right) \psi[\mathbf{K}_{\mathcal{L},\mathcal{L}^*}]\mathbf{q}^* \bigg],$$

*where*

$$\mathbf{w}_0 \triangleq \sum_{i=1}^{k^*} \mathbf{w}_i^*.$$

Replacing (11) in (9) gives us the loss function achieved at the local optimum. In order to simplify the loss expression, without loss of generality, from now on we replace $\mathbf{U}\mathbf{S}$ with $\mathbf{U}$ (note that there is essentially no difference between $\mathbf{U}_{\mathcal{L}}\mathbf{S}$ and $\mathbf{U}_{\mathcal{L}}$ as the columns of $\mathbf{U}_{\mathcal{L}}\mathbf{S}$ are the columns of $\mathbf{U}_{\mathcal{L}}$ with *adjusted* orientations.). Moreover, to simplify the analysis of this section, we make the following assumptions.

**Assumption 1** Recall that we assume that all $\mathbf{s}_i$ for $1 \leq i \leq r$ are equal to $\pm 1$. Our analysis extends naturally to other cases. Moreover, we assume that $\mathbf{w}_0 = 0$. This assumption has a negligible effect on our estimate of the value of the loss function achieved in the local minimum in many cases. For example, when $\mathbf{w}_i^*$ are i.i.d. $\mathcal{N}(0, (1/d)\mathbf{I})$ random vectors, $\mathbf{w}_0$ is a $\mathcal{N}(0, (r^*/d)\mathbf{I})$ random vector and therefore $\|\mathbf{w}_0\|_2 = \Theta(\sqrt{r^*})$. On the other hand, $\|\mathbf{q}^*\|_2 = \Theta(r^*)$. Hence, in the case where $r^*$ is large, the value of the loss function in the local minimum is controlled by the terms involving $\|\mathbf{q}^*\|_2^2$ in (9). Thus, we can ignore the terms involving $\mathbf{w}_0$ in this regime. Finally, we assume that $\psi[\mathbf{K}_{\mathcal{L}}]$ (and consequently $\mathbf{U}_{\mathcal{L}}^t\mathbf{U}_{\mathcal{L}} + \psi[\mathbf{K}_{\mathcal{L}}]$) is invertible.

**Theorem 6** *Under assumptions 1, in a local minimum of the mismatched PNN optimization, we have*

$$L(\mathbf{W}) = \frac{1}{4}(\mathbf{q}^*)^t \left( \widetilde{\psi}[\mathbf{K}]/\psi[\mathbf{K}_{\mathcal{L}}] \right) \mathbf{q}^*, \tag{12}$$

*where*

$$\widetilde{\psi}[\mathbf{K}] = \begin{bmatrix} \psi[\mathbf{K}_{\mathcal{L}}] + \mathbf{U}_{\mathcal{L}}^t\mathbf{U}_{\mathcal{L}} & \psi[\mathbf{K}_{\mathcal{L},\mathcal{L}^*}] \\ \psi[\mathbf{K}_{\mathcal{L},\mathcal{L}^*}]^t & \psi[\mathbf{K}_{\mathcal{L}^*}] \end{bmatrix}.$$

The matrix $\widetilde{\psi}[\mathbf{K}]$ has an extra term of $\mathbf{U}_{\mathcal{L}}^t\mathbf{U}_{\mathcal{L}}$ (i.e., the linear kernel) compared to the matrix $\psi[\mathbf{K}]$. The effect of this term is the price of converging to a local optimum in a bad region. In the following, we analysis this effect in the asymptotic regime where $r, d \to \infty$ while $r/d$ is fixed.

**Theorem 7** *Consider the asymptotic case where $r = \gamma d$, $r^* > d + 1$, $\gamma > 1$ and $r, r^*, d \to \infty$. Assume that $k^* = r^*$ underlying weight vectors $\mathbf{w}_i^* \in \mathbb{R}^d$ are chosen uniformly at random in $\mathbb{R}^d$ while the PNN is trained over $r$ lines drawn uniformly at random in $\mathbb{R}^d$. Under assumption 1, at local optima, with probability $1 - 2\exp(-\mu^2 d)$, we have*

$$L(\mathbf{W}) \leq \frac{1}{4}\left(1 - \frac{2}{\pi} + (1 + \sqrt{\gamma} + \mu)^2 \frac{r^*}{r}\right) \|\mathbf{q}^*\|_2^2,$$

*where $\mu > 1$ is a constant.*

Comparing asymptotic error bounds of Theorems MT-6 and 7, we observe that the extra PNN approximation error because of the convergence to a local minimum at a bad parameter region is reflected in the constant parameter $\mu$, which is negligible if $r^*$ is significantly smaller than $r$.

## 8   A Minimax Analysis of the Naive Nearest Line Approximation Approach

In this section, we show that every realizable function by a two-layer neural network (i.e., every $f \in \mathcal{F}$) can be approximated arbitrarily closely using a function described by a two-layer PNN (i.e., $\hat{f} \in \mathcal{F}_{\mathcal{L},\mathcal{G}}$). We start by the following lemma on the continuity of the ReLU function on the weight parameter:

**Lemma 3** *For the ReLU function $\phi(.)$, we have the following property*

$$|\phi\left(\langle \mathbf{w}_1, \mathbf{x}\rangle\right) - \phi\left(\langle \mathbf{w}_2, \mathbf{x}\rangle\right)| \leq \|\mathbf{w}_1 - \mathbf{w}_2\|_2 \|\mathbf{x}\|_2.$$

Recall that $\mathbf{u}_i$ is the unit norm vector over the line $i$. Let $\mathcal{U} = \{\mathbf{u}_1, \mathbf{u}_2, \ldots, \mathbf{u}_r\} \subseteq \mathbb{R}^d$. Denote the set $\mathcal{U}^- = \{-\mathbf{u}_1, -\mathbf{u}_2, \ldots, -\mathbf{u}_r\}$.

**Definition 1** *For $\delta \in [0, \pi/2]$, we call $\mathcal{U}$ an angular $\delta$-net of $\mathcal{W}$ if for every $\mathbf{w} \in \mathcal{W}$, there exists $\mathbf{u} \in \mathcal{U} \cup \mathcal{U}^-$ such that $\theta_{\mathbf{u},\mathbf{w}} \leq \delta$.*

The following lemma indicates the size required for $\mathcal{U}$ to be an angular $\delta$-net of the unit Euclidean sphere $S^{n-1}$.

**Lemma 4** *Let $\delta \in [0, \pi/2]$. For the unit Euclidean sphere $S^{n-1}$, there exists an angular $\delta$-net $\mathcal{U}$, with*

$$|\mathcal{U}| \leq \frac{1}{2}\left(1 + \frac{\sqrt{2}}{\sqrt{1 - \cos\delta}}\right)^n.$$

The following is a corollary of the previous lemma.

**Corollary 3** *Consider a two-layer neural network with $s$-sparse weights (i.e., $\mathcal{W}$ is the set of $s$-sparse vectors.). In this case, using lemma 4, $\mathcal{U}$ is an angular $\delta$-net of $\mathcal{W}$ with*

$$|\mathcal{U}| = \frac{1}{2}\binom{d}{s}\left(1 + \frac{\sqrt{2}}{\sqrt{1 - \cos\delta}}\right)^s.$$

*Furthermore, if we know the sparsity patterns of $k$ neurons in the network (i.e., if we know the network architecture), $\tilde{\mathcal{U}}$ is an angular $\delta$-net of $\mathcal{W}$ with*

$$|\tilde{\mathcal{U}}| \leq \frac{k}{2}\left(1 + \frac{\sqrt{2}}{\sqrt{1 - \cos\delta}}\right)^s.$$

In order to have a measure of how accurately a function in $\mathcal{F}$ can be approximated by a function in $\mathcal{F}_{\mathcal{L}}$, we have the following definition:

**Definition 2** *Define $\mathcal{R}\left(\mathcal{F}, \mathcal{F}_{\mathcal{L},\mathcal{G}}\right)$, the minimax risk of approximating a function in $\mathcal{F}$ by a function in $\mathcal{F}_{\mathcal{L},\mathcal{G}}$, as the following*

$$\mathcal{R}\left(\mathcal{F}_{\mathcal{L},\mathcal{G}}, \mathcal{F}\right) := \max_{f \in \mathcal{F}} \min_{\hat{f} \in \mathcal{F}_{\mathcal{L},\mathcal{G}}} \mathbb{E}\left|f(\mathbf{x}) - \hat{f}(\mathbf{x})\right|, \tag{13}$$

*where the expectation is over $\mathbf{x} \sim \mathcal{N}(0, \mathbf{I})$.*

The following theorem bounds this minimax risk where $\mathcal{U}$ is an angular $\delta$-net of $\mathcal{W}$.

**Theorem 8** *Assume that for all $\mathbf{w} \in \mathcal{W}$, $\|\mathbf{w}\|_2 \leq M$. Let $\mathcal{U}$ be an angular $\delta$-net of $\mathcal{W}$. The minimax risk of approximating a function in $\mathcal{F}$ with a function in $\mathcal{F}_{\mathcal{L},\mathcal{G}}$ defined in (13) can be written as*

$$\mathcal{R}\left(\mathcal{F}_{\mathcal{L},\mathcal{G}}, \mathcal{F}\right) \leq kM\sqrt{2d(1-\cos\delta)}.$$

The following is a corollary of Theorem 8 and Corollary 3.

**Corollary 4** *Let $\mathcal{F}$ be the set of realizable functions by a two-layer neural network with $s$-sparse weights. There exists a set $\mathcal{L}$ and a neuron-to-line mapping $\mathcal{G}$ such that*

$$\mathcal{R}\left(\mathcal{F}_{\mathcal{L},\mathcal{G}}, \mathcal{F}\right) \leq \delta,$$

*and*

$$|\mathcal{L}| \leq \frac{1}{2}\binom{d}{s}\left(1 + \frac{2kM\sqrt{d}}{\delta}\right)^s.$$

*Further, if we know the sparsity patterns of $k$ neurons in the network (i.e., the network architecture), then*

$$|\mathcal{L}| \leq \frac{k}{2}\left(1 + \frac{2kM\sqrt{d}}{\delta}\right)^s.$$

## 9    More Details On Numerical Experiments

All experiments were implemented in Python 2.7 using the TensorFlow package. We numerically simulate random PNNs in the mismatched case as described in Section MT-5. To enforce the PNN architecture, we project gradients along the directions of PNN lines before updating the weights. For example, if we consider $\mathbf{w}_i^{(0)}$ as the initial set of $d$ weights connecting hidden neuron $i$ to the $d$ inputs, then the final set of weights $\mathbf{w}_i^{(T)}$ need to lie on the same line as $\mathbf{w}_i^{(0)}$. To guarantee this, before applying gradient updates to $\mathbf{w}_i$, we first project them along $\mathbf{w}_i^{(0)}$.

For PNNs, we use $10 \leq k \leq 100$ hidden neurons. For each value of $k$, we perform 25 trials of the following:

1. Generate one set of true labels using a fully-connected two-layer network with $d = 15$ inputs and $k^* = 20$ hidden neurons. Generate 10,000 ground-truth training samples and 10,000 test samples using a set of randomly chosen weights.

2. Initialize $k/2$ random $d$-dimensional unit-norm weight vectors.

3. Assign each weight vector to two hidden neurons. For the first neuron, scale the vector by a random number sampled uniformly between 0 and 1. For the second neuron, scale the vector by a random number sampled uniformly between -1 and 0.

4. Train the network via stochastic gradient descent using batches of size 100, 100 training epochs, no momentum, and a learning rate of $10^{-3}$ which decays every epoch at a rate of 0.95 every 390 epochs.

5. Check to make sure that final weights lie along the same lines as initial weights. Ignore results if this is not the case due to numerical errors.

6. Repeat steps 2-5 10 times. Return the normalized MSE (i.e., MSE normalized by the $L_2$ norm of $y$) in the test set over different initializations.

## 10    Proofs

### 10.1    Preliminary Lemmas

**Lemma 5** *Let $\mathbf{x} \sim \mathcal{N}(0, \mathbf{I})$. We have*

$$\mathbb{E}\left[\mathbf{1}\{\mathbf{w}_1^t\mathbf{x} > 0, \mathbf{w}_2^t\mathbf{x} > 0\}\mathbf{x}\mathbf{x}^t\right] = \frac{\pi - \theta_{\mathbf{w}_1,\mathbf{w}_2}}{2\pi}\mathbf{I} + \frac{\sin\left(\theta_{\mathbf{w}_1,\mathbf{w}_1}\right)}{2\pi}\mathbf{M}(\mathbf{w}_1, \mathbf{w}_2), \qquad (14)$$

*where*

$$\mathbf{M}(\mathbf{w}_1, \mathbf{w}_2) \triangleq \frac{1}{\sin(\theta_{\mathbf{w}_1,\mathbf{w}_2})^2}(\mathbf{w}_1, \mathbf{w}_2)\begin{pmatrix} -\cos(\theta_{\mathbf{w}_1,\mathbf{w}_2}) & 1 \\ 1 & -\cos(\theta_{\mathbf{w}_1,\mathbf{w}_2}) \end{pmatrix}(\mathbf{w}_1, \mathbf{w}_2)^t. \quad (15)$$

Note that $\mathbf{M}(\mathbf{w}_1, \mathbf{w}_2)\mathbf{w}_1 = \frac{\|\mathbf{w}_1\|}{\|\mathbf{w}_2\|}\mathbf{w}_2$, $\mathbf{M}(\mathbf{w}_1, \mathbf{w}_2)\mathbf{w}_2 = \frac{\|\mathbf{w}_2\|}{\|\mathbf{w}_1\|}\mathbf{w}_1$, and $\mathbf{M}(\mathbf{w}_1, \mathbf{w}_2)\mathbf{v} = 0$ for every vector $\mathbf{v} \perp \text{span}(\mathbf{w}_1, \mathbf{w}_2)$.

**Lemma 6** *Let $x \sim \mathcal{N}(0,1)$. We have*

$$\mathbb{E}\left[\mathbf{1}\{w_1 x > 0, w_2 x > 0\}x^2\right] = \frac{1 + s(w_i)s(w_j)}{4}. \quad (16)$$

**Lemma 7** *Consider*

$$\mathbf{M} = \begin{bmatrix} \mathbf{A} & \mathbf{A} + \mathbf{\Delta}_1 \\ \mathbf{A}^t + \mathbf{\Delta}_1^t & \mathbf{A} + \mathbf{\Delta}_2 \end{bmatrix} \succeq 0$$

*where $\|\mathbf{\Delta}_1\|_2 \leq \sigma_1$, $\|\mathbf{\Delta}_2\|_2 \leq \sigma_2$ and $\lambda_{\min}(\mathbf{A}) \geq \delta$. Then*

$$\|\mathbf{M}/\mathbf{A}\|_2 \leq \frac{\sigma_1^2}{\delta} + 2\sigma_1 + \sigma_2.$$

**Proof 1** *Note that*

$$\begin{aligned} \mathbf{M}/\mathbf{A} &= \mathbf{A} + \mathbf{\Delta}_2 - \left(\mathbf{A} + \mathbf{\Delta}_1^t\right)\mathbf{A}^{-1}\left(\mathbf{A} + \mathbf{\Delta}_1\right) \\ &= \mathbf{A} + \mathbf{\Delta}_2 - \mathbf{A} - \mathbf{\Delta}_1^t - \mathbf{\Delta}_1 - \mathbf{\Delta}_1^t\mathbf{A}^{-1}\mathbf{\Delta}_1 \\ &= \mathbf{\Delta}_2 - \mathbf{\Delta}_1^t - \mathbf{\Delta}_1 - \mathbf{\Delta}_1^t\mathbf{A}^{-1}\mathbf{\Delta}_1. \end{aligned}$$

*Hence,*

$$\begin{aligned} \|\mathbf{M}/\mathbf{A}\|_2 &= \|\mathbf{\Delta}_2 - \mathbf{\Delta}_1^t - \mathbf{\Delta}_1 - \mathbf{\Delta}_1^t\mathbf{A}^{-1}\mathbf{\Delta}_1\|_2 \\ &\leq \|\mathbf{\Delta}_1^t\|_2\|\mathbf{A}^{-1}\|_2\|\mathbf{\Delta}_1\|_2 + 2\|\mathbf{\Delta}_1\|_2 + \|\mathbf{\Delta}_2\|_2 \\ &\leq \frac{\sigma_1^2}{\delta} + 2\sigma_1 + \sigma_2. \end{aligned}$$

**Lemma 8** *Suppose $\lambda_{min}(\mathbf{A}) \geq c > 0$ for some c. Then, for sufficiently small $\|\mathbf{\Delta}\|$, we have*

$$(\mathbf{A} + \mathbf{\Delta})^{-1} - \mathbf{A}^{-1} = \mathbf{A}^{-1}\tilde{\mathbf{\Delta}}\mathbf{A}^{-1} \quad (17)$$

*where $\|\tilde{\mathbf{\Delta}}\| \leq 2\|\mathbf{\Delta}\|$.*

**Proof 2** *From [30], we have*

$$(\mathbf{A} + \mathbf{\Delta})^{-1} = \mathbf{A}^{-1} - \mathbf{A}^{-1}\mathbf{\Delta}(\mathbf{I} + \mathbf{A}^{-1}\mathbf{\Delta})^{-1}\mathbf{A}^{-1}. \quad (18)$$

*Let*

$$\tilde{\mathbf{\Delta}} := -\mathbf{\Delta}(\mathbf{I} + \mathbf{A}^{-1}\mathbf{\Delta})^{-1}. \quad (19)$$

*Thus, we have*

$$\begin{aligned} \|\tilde{\mathbf{\Delta}}\| &\leq \|\mathbf{\Delta}\|\|(\mathbf{I} + \mathbf{A}^{-1}\mathbf{\Delta})^{-1}\| &(20) \\ &= \|\mathbf{\Delta}\|\frac{1}{\lambda_{min}(\mathbf{I} + \mathbf{A}^{-1}\mathbf{\Delta})}. \end{aligned}$$

*Moreover, if $\|\mathbf{\Delta}\| \leq c/2$, we have*

$$\lambda_{min}(\mathbf{I} + \mathbf{A}^{-1}\mathbf{\Delta}) \geq 1 - \|\mathbf{A}^{-1}\mathbf{\Delta}\| \quad (21)$$

$$\geq 1 - \frac{\|\mathbf{\Delta}\|}{\lambda_{min}(\mathbf{A})} \geq \frac{1}{2}. \quad (22)$$

*Using (20) and (21), for $\|\mathbf{\Delta}\| \leq c/2$, we have $\|\tilde{\mathbf{\Delta}}\| \leq 2\|\mathbf{\Delta}\|$. This completes the proof.*

**Lemma 9** *Let* $\mathbf{A} = \alpha_1 \mathbf{I}_n + \beta_1 \mathbf{1}_n$. *Then*

$$\mathbf{A}^{-1} = \alpha_2 \mathbf{I}_n + \beta_2 \mathbf{1}_n, \tag{23}$$

*where*

$$\alpha_2 = \frac{1}{\alpha_1} \tag{24}$$

$$\beta_2 = -\frac{-\beta_1}{\alpha_1^2 + \alpha_1 \beta_1 n}.$$

## 10.2  Proof of Theorem 1

In this case, we can re-write $L(\mathbf{W})$ as follows:

$$L(\mathbf{W}) = \mathbb{E}\left[\left(\sum_{i=1}^{k} \mathbf{1}\{w_i x > 0\} w_i x - \sum_{i=1}^{k} \mathbf{1}\{w_i^* x > 0\} w_i^* x\right)^2\right] \tag{25}$$

$$= \mathbb{E}\left[\left(\sum_{i=1}^{k} \mathbf{1}\{w_i x > 0\} w_i x\right)^2\right] + \mathbb{E}\left[\left(\sum_{i=1}^{k} \mathbf{1}\{w_i^* x > 0\} w_i^* x\right)^2\right]$$

$$- \mathbb{E}\left[\sum_{i,j} \mathbf{1}\{w_i x > 0, w_j^* x > 0\} w_i w_j^* x^2\right].$$

The first term of (25) can be simplified as follows:

$$\mathbb{E}\left[\left(\sum_{i=1}^{k} \mathbf{1}\{w_i x > 0\} w_i x\right)^2\right] = \frac{1}{2}\sum_{i=1}^{k} w_i^2 + \frac{1}{4}\sum_{i \neq j} w_i w_j \left(s(w_i)s(w_j) + 1\right) \tag{26}$$

$$= \frac{1}{4}\left(\sum_{i=1}^{k} w_i^2 + \sum_{i \neq j} w_i w_j\right) + \frac{1}{4}\left(\sum_{i=1}^{k} s(w_i) w_i^2 + \sum_{i \neq j} s(w_i)s(w_j) w_i w_j\right)$$

$$= \frac{1}{4}\left(\sum_{i=1}^{k} w_i\right)^2 + \frac{1}{4}\left(\sum_{i=1}^{k} s(w_i) w_i\right)^2,$$

where the first step follows from Lemma 6. The second term of (25) can be simplified similarly. The third term of (25) can be re-written as

$$\mathbb{E}\left[\sum_{i,j} \mathbf{1}\{w_i x > 0, w_j^* x > 0\} w_i w_j^* x^2\right] = \frac{1}{4}\sum_{i,j} w_i w_j^* (s(w_i)s(w_j^*) + 1) \tag{27}$$

$$= \frac{1}{4}\left(\sum_{i,j} w_i w_j^*\right) + \frac{1}{4}\left(\sum_{i,j} s(w_i)s(w_j^*) w_i w_j^*\right). \tag{28}$$

Substituting (26) and (27) in (25), we have

$$L(\mathbf{W}) = \frac{1}{4}\left((\sum_{i=1}^{k} w_i)^2 + (\sum_{i=1}^{k} w_i^*)^2 - (\sum_{i,j} w_i w_j^*)\right)^2 \tag{29}$$

$$+ \frac{1}{4}\left((\sum_{i=1}^{k} s(w_i) w_i)^2 + (\sum_{i=1}^{k} s(w_i^*) w_i^*)^2 - (\sum_{i,j} s(w_i)s(w_j^*) w_i w_j^*)\right)^2$$

$$= \frac{1}{4}\left(\sum_{i=1}^{k} w_i - \sum_{i=1}^{k} w_i^*\right)^2 + \frac{1}{4}\left(\sum_{i=1}^{k} s(w_i) w_i - \sum_{i=1}^{k} s(w_i^*) w_i^*\right)^2. \tag{30}$$

Therefore, $L(\mathbf{W}) = 0$ if and only if $\sum_{i=1}^{k} w_i = \sum_{i=1}^{k} w_i^*$ and $\sum_{i=1}^{k} s(w_i) w_i = \sum_{i=1}^{k} s(w_i^*) w_i^*$.
This completes the proof.

## 10.3 Proof of Theorem 2

First, we characterize the gradient of the loss function with respect to $w_j$:

$$\nabla_{w_j} L(\mathbf{W}) = 2\mathbb{E}\left[\left(\sum_{i=1}^{k} \mathbf{1}\{w_i x > 0\}w_i x - \sum_{i=1}^{k} \mathbf{1}\{w_i^* x > 0\}w_i^* x\right)\left(\mathbf{1}\{w_j x > 0\}x\right)\right] \quad (31)$$

$$= \frac{1}{2}\sum_{i=1}^{k} w_i w_j \left(1 + s(w_i)s(w_j)\right) - \frac{1}{2}\sum_{i=1}^{k} w_i^* w_j \left(1 + s(w_i^*)s(w_j)\right)$$

$$= \frac{1}{2}\left(\sum_{i=1}^{k} w_i - \sum_{i=1}^{k} w_i^*\right) + \frac{s(w_j)}{2}\left(\sum_{i=1}^{k} s(w_i)w_i - \sum_{i=1}^{k} s(w_i^*)w_i^*\right), \quad (32)$$

where the first step follows from Lemma 6. A necessary condition to have $\mathbf{W}$ as a local optimizer is $\nabla_{w_j} L(\mathbf{w}) = 0$ for every $j$.

Consider a region $R(\mathbf{s})$ where $\mathbf{s} \neq \pm\mathbf{1}$. Thus, there are two indices $j_1$ and $j_2$ such that $s(w_{j_1}) > 0$ and $s(w_{j_2}) < 0$. To have a local optimizer in this region, we need to have

$$\left(\sum_{i=1}^{k} w_i - \sum_{i=1}^{k} w_i^*\right) + s(w_{j_1})\left(\sum_{i=1}^{k} s(w_i)w_i - \sum_{i=1}^{k} s(w_i^*)w_i^*\right) = 0, \quad (33)$$

$$\left(\sum_{i=1}^{k} w_i - \sum_{i=1}^{k} w_i^*\right) + s(w_{j_2})\left(\sum_{i=1}^{k} s(w_i)w_i - \sum_{i=1}^{k} s(w_i^*)w_i^*\right) = 0.$$

Summing these two equations leads to the following conditions:

$$\sum_{i=1}^{k} w_i - \sum_{i=1}^{k} w_i^* = 0, \quad (34)$$

$$\sum_{i=1}^{k} s(w_i)w_i - \sum_{i=1}^{k} s(w_i^*)w_i^* = 0.$$

On the other hand, Theorem 1 indicates that if $\mathbf{W}$ satisfies these conditions, its loss value is equal to zero. Thus, such local optimizers are global optimizers. In regions $R(\pm\mathbf{1})$, to have $\nabla_{w_j} L(\mathbf{W}) = 0$ for every $j$, we only need to have the condition $\sum_{i=1}^{k} w_i - \sum_{i=1}^{k} w_i^* = 0$. In this case, if $s(\mathbf{W}^*) \neq \pm\mathbf{1}$, we will have bad local optimizers. This completes the proof.

## 10.4 Proof of Theorem 3

For every $1 \leq i, j \leq k$, we have

$$\nabla^2_{w_i, w_j} L(\mathbf{W}) = 2\mathbb{E}[\mathbf{1}\{w_i x > 0, w_j x > 0\}x^2] = \frac{s(w_i)s(w_j)}{2} \quad (35)$$

Let $\mathbf{H}$ be the Hessian matrix where $\mathbf{H}(i, j) = \nabla^2_{w_i, w_j} L(\mathbf{W})$. Thus, in the region $R(\mathbf{s})$, we have

$$\mathbf{H} = \frac{1}{2}\mathbf{1} + \frac{1}{2}\mathbf{s}\mathbf{s}^t. \quad (36)$$

Note that $\mathbf{H}$ is positive semidefinite and its rank is equal to two except when $\mathbf{s} = \pm\mathbf{1}$ in which case its rank is equal to one.

## 10.5 Proof of Theorem 4

We can re-write $L(\mathbf{W})$ as follows:

$$L(\mathbf{W}) = \mathbb{E}\left[\left(\sum_{i=1}^{k}\mathbf{1}\{\mathbf{w}_i^t\mathbf{x} > 0\}\mathbf{w}_i^t\mathbf{x} - \sum_{i=1}^{k}\mathbf{1}\{(\mathbf{w}_i^*)^t\mathbf{x} > 0\}(\mathbf{w}_i^*)^t\mathbf{x}\right)^2\right] \tag{37}$$

$$= \mathbb{E}\left[\left(\sum_{i=1}^{k}\mathbf{1}\{\mathbf{w}_i^t\mathbf{x} > 0\}\mathbf{w}_i^t\mathbf{x}\right)^2\right] + \mathbb{E}\left[\left(\sum_{i=1}^{k}\mathbf{1}\{(\mathbf{w}_i^*)^t\mathbf{x} > 0\}(\mathbf{w}_i^*)^t\mathbf{x}\right)^2\right]$$

$$- 2\mathbb{E}\left[\sum_{i,j}\mathbf{1}\{\mathbf{w}_i^t\mathbf{x} > 0, (\mathbf{w}_j^*)^t\mathbf{x} > 0\}(\mathbf{w}_i^t\mathbf{x})((\mathbf{w}_j^*)^t\mathbf{x})\right].$$

The first term can be re-written as

$$\mathbb{E}\left[\left(\sum_{i=1}^{k}\mathbf{1}\{\mathbf{w}_i^t\mathbf{x} > 0\}\mathbf{w}_i^t\mathbf{x}\right)^2\right] = \frac{1}{2}\sum_{i=1}^{k}w_i^2 + \frac{1}{4}\sum_{\substack{i\neq j \\ g(i)=g(j)}}(w_iw_j + |w_i||w_j|) \tag{38}$$

$$+ \frac{1}{2\pi}\sum_{\substack{i,j \\ g(i)\neq g(j)}}|w_i||w_j|$$

where the first step follows from Lemma 5. A similar equation can be written for the second term of (37). The third term of (37) can be re-written as

$$-2\mathbb{E}\left[\sum_{i,j}\mathbf{1}\{\mathbf{w}_i^t\mathbf{x} > 0, (\mathbf{w}_j^*)^t\mathbf{x} > 0\}(\mathbf{w}_i^t\mathbf{x})\left((\mathbf{w}_j^*)^t\mathbf{x}\right)\right] = -\frac{1}{2}\sum_{\substack{i,j \\ g(i)=g(j)}}(w_iw_j^* + |w_i||w_j^*|) \tag{39}$$

$$- \frac{1}{\pi}\sum_{\substack{i,j \\ g(i)\neq g(j)}}|w_i||w_j^*|$$

Substituting (38) and (39) in (37) we have

$$4L(\mathbf{W}) = \sum_{r=1}^{d}\left(\sum_{i\in\mathcal{G}_r}w_i - w_i^*\right)^2 + \sum_{r=1}^{d}(q_r - q_r^*)^2 + \frac{2}{\pi}\sum_{r\neq t}(q_r - q_r^*)(q_t - q_t^*). \tag{40}$$

This completes the proof.

## 10.6 Proof of Theorem 5

First, we characterize the gradient of the loss function with respect to $\mathbf{w}_j$:

$$\nabla_{\mathbf{w}_j}L(\mathbf{W}) = 2\mathbb{E}\left[(\mathbf{1}\{\mathbf{w}_j^t\mathbf{x} > 0\}\mathbf{x})\left(\sum_{i=1}^{k}\mathbf{1}\{\mathbf{w}_i^t\mathbf{x} > 0\}\mathbf{w}_i^t\mathbf{x} - \sum_{i=1}^{k}\mathbf{1}\{(\mathbf{w}_i^*)^t\mathbf{x} > 0\}(\mathbf{w}_i^*)^t\mathbf{x}\right)\right]$$

$$\tag{41}$$

$$= \frac{1}{2}\sum_{\substack{i \\ g(i)=g(j)}}(1 + s(w_i)s(w_j))\mathbf{w}_i + \frac{1}{2}\sum_{\substack{i \\ g(i)\neq g(j)}}\left(\mathbf{w}_i + \frac{2\|\mathbf{w}_i\|}{\pi\|\mathbf{w}_j\|}\mathbf{w}_j\right)$$

$$- \frac{1}{2}\sum_{\substack{i \\ g(i)=g(j)}}(1 + s(w_i^*)s(w_j))\mathbf{w}_i^* - \frac{1}{2}\sum_{\substack{i \\ g(i)\neq g(j)}}\left(\mathbf{w}_i^* + \frac{2\|\mathbf{w}_i^*\|}{\pi\|\mathbf{w}_j\|}\mathbf{w}_j\right)$$

$$= \frac{1}{2}\left(\sum_{i=1}^{k}\mathbf{w}_i - \mathbf{w}_i^*\right) + \frac{s(w_j)}{2}\left((q_{g(j)} - q_{g(j)}^*) + \frac{2}{\pi}\sum_{r\neq g(j)}(q_r - q_r^*)\right)\mathbf{e}_{g(j)}$$

where the first step follows from Lemma 5. A necessary condition to have $\mathbf{W}$ as a local optimizer of optimization MT-(6) is that the projection gradient is zero for every $j$, i.e., $< \bigtriangledown_{\mathbf{w}_j} L(\mathbf{W}), \mathbf{e}_{g(j)} > = 0$ for every $j$.

Under the condition of Theorem 5, for every $1 \le r \le d$, there exists $j_1 \ne j_2 \in \mathcal{G}_r$ such that $s(\mathbf{w}_{j_1})s(\mathbf{w}_{j_2}) = -1$. Thus, summing up (41) for $j_1$ and $j_2$, we have

$$\mathbf{e}_r^t \left( \sum_{i=1}^{k} \mathbf{w}_i - \mathbf{w}_i^* \right) = 0. \tag{42}$$

Since this is true for every $1 \le r \le d$, we have $\sum_{i=1}^{k} \mathbf{w}_i - \mathbf{w}_i^* = 0$. The second term of (41) is a vector with a non-zero element at its $g(j)$ component. Having the first term of (41) equal to zero, the second term should be zero in local optimizers. This leads to the set of equations

$$\mathbf{C}(\mathbf{q} - \mathbf{q}^*) = 0 \tag{43}$$

where $\mathbf{C}$ is defined in (4). On the other hand, using Theorem 4, having these conditions lead to $L(\mathbf{W}) = 0$. In other words, under the conditions of Theorem 5, every local optimizer is a global optimizer for a one-degree PNN. This completes the proof.

## 10.7  Proof of Theorem MT-1

First, we decompose $L(\mathbf{W})$ to three terms similar to (37). Then the first term can be re-written as follows:

$$\mathbb{E}\left[ \left( \sum_{i=1}^{k} \mathbf{1}\{\mathbf{w}_i^t\mathbf{x} > 0\}\mathbf{w}_i^t\mathbf{x} \right)^2 \right] \tag{44}$$

$$= \frac{1}{2}\sum_{i=1}^{k} \|\mathbf{w}_i\|^2 + \sum_{l=1}^{r} \sum_{\substack{i \ne j \\ i,j \in \mathcal{G}_l}} \frac{1 + s(\mathbf{w}_i)s(\mathbf{w}_j)}{4} \|\mathbf{w}_i\|\|\mathbf{w}_j\|$$

$$+ \sum_{l \ne l'} \sum_{\substack{i \in \mathcal{G}_l \\ j \in \mathcal{G}_{l'}}} \left( \frac{(\pi - \theta_{\mathbf{w}_i,\mathbf{w}_j}\cos(\theta_{\mathbf{w}_i,\mathbf{w}_j})) + \sin(\theta_{\mathbf{w}_i,\mathbf{w}_j})}{2\pi} \right) \|\mathbf{w}_i\|\|\mathbf{w}_j\|$$

$$= \frac{1}{2}\sum_{i=1}^{k} \|\mathbf{w}_i\|^2 + \sum_{l=1}^{r} \sum_{\substack{i \ne j \\ i,j \in \mathcal{G}_l}} \frac{1 + s(\mathbf{w}_i)s(\mathbf{w}_j)}{4} \|\mathbf{w}_i\|\|\mathbf{w}_j\|$$

$$+ \frac{1}{2\pi}\sum_{l \ne l'} \sum_{\substack{i \in \mathcal{G}_l \\ j \in \mathcal{G}_{l'}}} \left( s(\mathbf{w}_i)s(\mathbf{w}_j)\cos(\mathbf{A}_\mathcal{L}(l,l')) \left( \frac{\pi}{2} - \left( \mathbf{A}_\mathcal{L}(l,l') - \frac{\pi}{2} \right)s(\mathbf{w}_i)s(\mathbf{w}_j) \right) + \sin(\mathbf{A}_\mathcal{L}(l,l')) \right) \|\mathbf{w}_i\|\|\mathbf{w}_j\|$$

$$= \frac{1}{4}\left( \sum_{i=1}^{k} \|\mathbf{w}_i\|^2 + \sum_{i \ne j} <\mathbf{w}_i,\mathbf{w}_j> \right) + \frac{1}{4}\left( \sum_{i=1}^{k} \|\mathbf{w}_i\|^2 + \sum_{i \ne j} \mathbf{A}_\mathcal{L}(g(i),g(j))\|\mathbf{w}_i\|\|\mathbf{w}_j\| \right)$$

where the first step follows from Lemma 5, and in the second step, we use

$$\theta_{\mathbf{w}_i,\mathbf{w}_j} = \frac{\pi}{2} + (a_{g(i),g(j)} - \frac{\pi}{2})s(\mathbf{w}_i)s(\mathbf{w}_j). \tag{45}$$

A similar argument can be mentioned for the second term of (37). The third term of (37) can be re-written as

$$-2\mathbb{E}\left[\sum_{i,j}\mathbf{1}\{\mathbf{w}_i^t\mathbf{x}>0,(\mathbf{w}_j^*)^t\mathbf{x}>0\}(\mathbf{w}_i^t\mathbf{x})\left((\mathbf{w}_j^*)^t\mathbf{x}\right)\right] = -\frac{1}{2}\sum_{l=1}^{r}\sum_{\substack{i,j\\i,j\in\mathcal{G}_l}}(1+s(\mathbf{w}_i)s(\mathbf{w}_j^*))\|\mathbf{w}_i\|\|\mathbf{w}_j\|$$

$$(46)$$

$$+\sum_{l\neq l'}\sum_{\substack{i\in\mathcal{G}_l\\j\in\mathcal{G}_{l'}}}<\mathbf{w}_i,\mathbf{w}_j^*>+\mathbf{A}_{\mathcal{L}}(l,l')\|\mathbf{w}_i\|\|\mathbf{w}_j^*\|$$

$$=-\frac{1}{2}\sum_{i,j}<\mathbf{w}_i,\mathbf{w}_j^*>+\mathbf{A}_{\mathcal{L}}(g(i),g(j))\|\mathbf{w}_i\|\|\mathbf{w}_j^*\|$$

where we use Lemma 5 and equation (37). Substituting (44) and (46) in (37) completes the proof.

### 10.8 Proof of Lemma MT-1

Note that the matrix $\mathbf{K}=\cos[\mathbf{A}_{\mathcal{L}}]$ is a covariance matrix and thus is positive semidefinite. For the function $\psi(.)$ defined as in MT-(8), we have

$$\frac{\partial^j\psi}{\partial x^j}=\begin{cases}0, & \text{if j is odd}\\ \frac{2/\pi\prod_{i=1}^{j-2}(2i-1)}{2^{j-2}}, & \text{if j is even}\end{cases}\qquad(47)$$

Thus, for every $j\geq 1$, we have $\frac{\partial^j\psi}{\partial x^j}\geq 0$. Using Theorem 4.1 (i) of reference [31] completes the proof.

### 10.9 Proof of Theorem MT-2

We characterize the gradient of the loss function with respect to $\mathbf{w}_j$:

$$\nabla_{\mathbf{w}_j}L(\mathbf{w})=2\mathbb{E}\left[\left(\mathbf{1}\{\mathbf{w}_j^t\mathbf{x}>0\}\mathbf{x}\right)\left(\sum_{i=1}^{k}\mathbf{1}\{\mathbf{w}_i^t\mathbf{x}>0\}\mathbf{w}_i^t\mathbf{x}-\sum_{i=1}^{k}\mathbf{1}\{(\mathbf{w}_i^*)^t\mathbf{x}>0\}(\mathbf{w}_i^*)^t\mathbf{x}\right)\right]$$

$$(48)$$

$$=\sum_{l=1}^{r}\left(\sum_{i\in\mathcal{G}_l}\left(\frac{\pi-\theta_{\mathbf{w}_i,\mathbf{w}_j}}{2\pi}\mathbf{I}+\frac{\sin(\theta_{\mathbf{w}_i,\mathbf{w}_j})}{2\pi}\mathbf{M}(\mathbf{w}_i,\mathbf{w}_j)\right)\mathbf{w}_i\right.$$

$$-\left.\left(\frac{\pi-\theta_{\mathbf{w}_i^*,\mathbf{w}_j}}{2\pi}\mathbf{I}+\frac{\sin(\theta_{\mathbf{w}_i^*,\mathbf{w}_j})}{2\pi}\mathbf{M}(\mathbf{w}_i^*,\mathbf{w}_j)\right)\mathbf{w}_i^*\right)$$

$$=\frac{1}{4}\sum_{i=1}^{k}(\mathbf{w}_i-\mathbf{w}_i^*)+s(\mathbf{w}_j)\left(\sum_{l=1}^{r}\sum_{i\in\mathcal{G}_l}\frac{(\pi/2-\mathbf{A}_{\mathcal{L}}(l,g(j)))(\|\mathbf{w}_i\|-\|\mathbf{w}_i^*\|)}{2\pi}\mathbf{u}_l\right.$$

$$+\left.\frac{\sin(\mathbf{A}_{\mathcal{L}}(l,g(j)))(\|\mathbf{w}_i\|-\|\mathbf{w}_i^*\|)}{2\pi}\mathbf{u}_{g(i)}\right)$$

where the first step follows from Lemma 5, and in the second step, we use (45).

A necessary condition to have $\mathbf{W}$ as a local optimizer is that the projected gradient is zero for every $j$, i.e., $\mathbf{u}_{g(j)}^t\nabla_{\mathbf{w}_j}L(\mathbf{W})=0$ for every $j$. Under the conditions of Theorem MT-2, over $d$ distinct lines, there exists $j_1\neq j_2\in\mathcal{G}_r$ such that $s(\mathbf{w}_{j_1})s(\mathbf{w}_{j_2})=-1$. Thus, summing up (48) for $j_1$ and $j_2$, we have

$$\mathbf{u}_r^t\left(\sum_{i=1}^{k}\mathbf{w}_i-\mathbf{w}_i^*\right)=0.\qquad(49)$$

Since this is true for $d$ distinct and thus linearly independent lines, we have $\sum_i \mathbf{w}_i - \mathbf{w}_i^* = 0$. Therefore, the inner product of the second term of (48) with $\mathbf{u}_{g(j)}$ should be zero in local optimizers. This leads to the following equation:

$$\sum_{k=1}^{r} \sum_{i \in \mathcal{G}_l} \psi[\mathbf{K}_{\mathcal{L}}](l, g(j)) \left(\|\mathbf{w}_i\| - \|\mathbf{w}_i^*\|\right) = \sum_{r=1}^{l} \psi[\mathbf{K}_{\mathcal{L}}](l, g(j)) (q_l - q_l^*) = 0. \quad (50)$$

Since this should hold for every $j$, a necessary condition for $\mathbf{W}$ to be a local optimizer is $\psi[\mathbf{K}_{\mathcal{L}}](\mathbf{q} - \mathbf{q}^*) = 0$. On the other hand, using Theorem MT-1, such conditions lead to having $L(\mathbf{W}) = 0$. Therefore, such local optimizers are global optimizers. This completes the proof.

## 10.10 Proof of Theorem MT-3

The proof is similar to the one of Theorem MT-1.

## 10.11 Proof of Theorem MT-4

A necessary condition for a point to be a local optimizer is that $\mathbf{u}_{g(j)}^t \nabla_{\mathbf{w}_j} L(\mathbf{W}) = 0$ for every $j$. Similarly to the proof of Theorem MT-2, under the condition of Theorem MT-4, we have $\sum_{i=1}^{k} \mathbf{w}_i - \sum_{i=1}^{k^*} \mathbf{w}_i^* = 0$. This leads to the following equation in local optimizers:

$$\psi[\mathbf{K}_{\mathcal{L}}]\mathbf{q} = \psi[\mathbf{K}_{\mathcal{L},\mathcal{L}^*}]\mathbf{q}^*. \quad (51)$$

Replacing this equation in the loss function completes the proof.

## 10.12 Proof of Lemma 1

To simplify notations, define

$$\mathbf{D} = \psi[\mathbf{K}] = \begin{bmatrix} \mathbf{D}_{11} & \mathbf{D}_{12} \\ \mathbf{D}_{12}^t & \mathbf{D}_{22} \end{bmatrix} \succeq 0$$

Note that since $\psi(.)$ has Lipschitz constant $L \leq 1$, we have

$$\left|(\mathbf{D}_{22} - \mathbf{D}_{11})_{ij}\right| \leq \left|\left((\mathbf{U}^*)^t \mathbf{U}^* - \mathbf{U}^t \mathbf{U}\right)_{ij}\right|$$
$$= \left|\left((\mathbf{U} + \mathbf{Z})^t (\mathbf{U} + \mathbf{Z}) - \mathbf{U}^t \mathbf{U}\right)_{ij}\right| = \left|\left(\mathbf{U}^t \mathbf{Z} + \mathbf{Z}^t \mathbf{U} + \mathbf{Z}^t \mathbf{Z}\right)_{ij}\right|$$
$$\leq \|\mathbf{U}_{.,i}\|_2 \|\mathbf{Z}_{.,j}\|_2 + \|\mathbf{U}_{.,j}\|_2 \|\mathbf{Z}_{.,i}\|_2 + \|\mathbf{Z}_{.,i}\|_2 \|\mathbf{Z}_{.,j}\|_2$$
$$\leq \|\mathbf{Z}_{.,j}\|_2 + \|\mathbf{Z}_{.,i}\|_2 + \|\mathbf{Z}_{.,i}\|_2 \|\mathbf{Z}_{.,j}\|_2,$$

where the last step follows from the fact that $\|\mathbf{U}_{.,i}\| = 1$. Hence,

$$\|\mathbf{D}_{22} - \mathbf{D}_{11}\|_2 \leq \|\mathbf{D}_{22} - \mathbf{D}_{11}\|_F \leq 2\sqrt{r}\|\mathbf{Z}\|_F + \|\mathbf{Z}\|_F^2. \quad (52)$$

Similarly,

$$\left|(\mathbf{D}_{12} - \mathbf{D}_{11})_{ij}\right| \leq \left|\left(\mathbf{U}^t \mathbf{U}^* - \mathbf{U}^t \mathbf{U}\right)_{ij}\right|$$
$$= \left|\left(\mathbf{U}^t (\mathbf{U} + \mathbf{Z}) - \mathbf{U}^t \mathbf{U}\right)_{ij}\right| = \left|\left(\mathbf{U}^t \mathbf{Z}\right)_{ij}\right|$$
$$\leq \|\mathbf{U}_{.,i}\|_2 \|\mathbf{Z}_{.,j}\|_2 \leq \|\mathbf{Z}_{.,j}\|_2.$$

Thus,

$$\|\mathbf{D}_{12} - \mathbf{D}_{11}\|_2 \leq \|\mathbf{D}_{12} - \mathbf{D}_{11}\|_F \leq \sqrt{r}\|\mathbf{Z}\|_F. \quad (53)$$

Further, note that using (52),

$$\lambda_{\min}(\mathbf{D}_{11}) \geq \lambda_{\min}(\mathbf{D}_{22}) - \|\mathbf{D}_{22} - \mathbf{D}_{11}\|_2 \geq \delta - 2\sqrt{r}\|\mathbf{Z}\|_F - \|\mathbf{Z}\|_F^2 \geq \frac{\delta}{2}, \quad (54)$$

under the assumptions of the Lemma. Hence, combining (52), (53), (54), using Lemma 7, we have

$$\|\mathbf{D}/\mathbf{D}_{11}\|_2 \leq \frac{2\|\mathbf{D}_{12} - \mathbf{D}_{11}\|_2^2}{\delta} + 2\|\mathbf{D}_{12} - \mathbf{D}_{11}\|_2 + \|\mathbf{D}_{22} - \mathbf{D}_{11}\|_2$$
$$\leq \left(1 + \frac{2r}{\delta}\right)\|\mathbf{Z}\|_F^2 + 4\sqrt{r}\|\mathbf{Z}\|_F.$$

## 10.13 Proof of Theorem MT-5

To simplify notations, we define

$$\mathbf{D} = \psi[\mathbf{K}_{\text{new}}] = \psi\left[\begin{pmatrix} 1 & \mathbf{z}_1 & \mathbf{z}_2 \\ \mathbf{z}_1^t & \mathbf{K}_{\mathcal{L}} & \mathbf{K}_{\mathcal{L},\mathcal{L}^*} \\ \mathbf{z}_2^t & \mathbf{K}_{\mathcal{L},\mathcal{L}^*}^t & \mathbf{K}_{\mathcal{L}^*} \end{pmatrix}\right] = \begin{pmatrix} 1 & \zeta_1^t & \zeta_2^t \\ \zeta_1 & \mathbf{D}_{11} & \mathbf{D}_{12} \\ \zeta_2 & \mathbf{D}_{12}^t & \mathbf{D}_{22} \end{pmatrix}$$

and

$$\mathbf{R}_1 = \mathbf{D}_{22} - \mathbf{D}_{12}^t \mathbf{D}_{11}^{-1} \mathbf{D}_{12} \ ,$$

$$\mathbf{R}_2 = \mathbf{D} \Bigg/ \begin{bmatrix} 1 & \zeta_1^t \\ \zeta_1 & \mathbf{D}_{11} \end{bmatrix}.$$

Note that since $\mathbf{D}$ is positive semidefinite (Lemma MT-1), we have

$$\begin{bmatrix} 1 & \zeta_1^t \\ \zeta_1 & \mathbf{D}_{11} \end{bmatrix} \succeq 0.$$

Hence

$$\begin{bmatrix} 1 & \zeta_1^t \\ \zeta_1 & \mathbf{D}_{11} \end{bmatrix} \Bigg/ \mathbf{D}_{11} = 1 - \langle \zeta_1, \mathbf{D}_{11}^{-1}\zeta_1 \rangle \geq 0.$$

We have

$$\mathbf{R}_2 = \mathbf{D}_{22} - \begin{bmatrix} \zeta_2 & \mathbf{D}_{12}^t \end{bmatrix} \begin{bmatrix} 1 & \zeta_1^t \\ \zeta_1 & \mathbf{D}_{11} \end{bmatrix}^{-1} \begin{bmatrix} \zeta_2^t \\ \mathbf{D}_{12} \end{bmatrix}$$

$$= \mathbf{D}_{11} - \begin{bmatrix} \zeta_2 & \mathbf{D}_{12}^t \end{bmatrix} \begin{bmatrix} \left(1 - \langle \zeta_1, \mathbf{D}_{11}^{-1}\zeta_1 \rangle\right)^{-1} & -\zeta_1^t \mathbf{D}_{11}^{-1}\left(1 - \langle \zeta_1, \mathbf{D}_{11}^{-1}\zeta_1 \rangle\right)^{-1} \\ -\mathbf{D}_{11}^{-1}\zeta_1 \left(1 - \langle \zeta_1, \mathbf{D}_{11}^{-1}\zeta_1 \rangle\right)^{-1} & \left(\mathbf{D}_{11} - \zeta_1\zeta_1^t\right)^{-1} \end{bmatrix} \begin{bmatrix} \zeta_2^t \\ \mathbf{D}_{12} \end{bmatrix}$$

$$= \mathbf{D}_{22} - \begin{bmatrix} \zeta_2 & \mathbf{D}_{12}^t \end{bmatrix} \begin{bmatrix} \left(1 - \langle \zeta_1, \mathbf{D}_{11}^{-1}\zeta_1 \rangle\right)^{-1}\left(\zeta_1^t - \zeta_1^t \mathbf{D}_{11}^{-1} \mathbf{D}_{12}\right) \\ -\left(1 - \langle \zeta_1, \mathbf{D}_{11}^{-1}\zeta_1 \rangle\right)^{-1}\mathbf{D}_{11}^{-1}\zeta_1\zeta_2^t + \left(\mathbf{D}_{11} - \zeta_1\zeta_1^t\right)^{-1}\mathbf{D}_{12} \end{bmatrix}$$

$$= \mathbf{D}_{22} + \left(1 - \langle \zeta_1, \mathbf{D}_{11}^{-1}\zeta_1 \rangle\right)^{-1}\left[-\zeta_2\zeta_2^t + \zeta_2\zeta_1^t \mathbf{D}_{11}^{-1} \mathbf{D}_{12} + \mathbf{D}_{12}^t \mathbf{D}_{11}^{-1}\zeta_1\zeta_2^t\right] - \mathbf{D}_{12}^t \left(\mathbf{D}_{11} - \zeta_1\zeta_1^t\right)^{-1}\mathbf{D}_{12}.$$

Using the Sherman-Morisson formula, we have

$$\left(\mathbf{D}_{11} - \zeta_1\zeta_1^t\right)^{-1} = \mathbf{D}_{11}^{-1} + \left(1 - \langle \zeta_1, \mathbf{D}_{11}^{-1}\zeta_1 \rangle\right)^{-1}\mathbf{D}_{11}^{-1}\zeta_1\zeta_1^t \mathbf{D}_{11}^{-1}.$$

Hence,

$$\mathbf{R}_2 = \mathbf{D}_{22} - \mathbf{D}_{12}^t \mathbf{D}_{11}^{-1}\mathbf{D}_{12} - \left(1 - \langle \zeta_1, \mathbf{D}_{11}^{-1}\zeta_1 \rangle\right)^{-1}\left[\zeta_2\zeta_2^t - \zeta_2\zeta_1^t \mathbf{D}_{11}^{-1}\mathbf{D}_{12} - \mathbf{D}_{12}^t \mathbf{D}_{11}^{-1}\zeta_1\zeta_2^t + \mathbf{D}_{12}^t \mathbf{D}_{11}^{-1}\zeta_1\zeta_1^t \mathbf{D}_{11}^{-1}\mathbf{D}_{12}\right]$$

$$= \mathbf{R}_1 - \left(1 - \langle \zeta_1, \mathbf{D}_{11}^{-1}\zeta_1 \rangle\right)^{-1}\left[\left(\zeta_2 - \mathbf{D}_{12}^t \mathbf{D}_{11}^{-1}\zeta_1\right)\left(\zeta_2 - \mathbf{D}_{12}^t \mathbf{D}_{11}^{-1}\zeta_1\right)^t\right]$$

$$= \mathbf{R}_1 - \alpha\mathbf{v}\mathbf{v}^t$$

where $\alpha \geq 0$, $\mathbf{v}$ are defined in the theorem. Hence, $\mathbf{R}_1 \succeq \mathbf{R}_2$ and $\|\mathbf{R}_1\|_2 \geq \|\mathbf{R}_2\|_2$. This completes the proof.

## 10.14 Proof of Theorem MT-6

To simplify notations, define

$$\mathbf{D} = \psi[\mathbf{K}] = \begin{bmatrix} \mathbf{D}_{11} & \mathbf{D}_{12} \\ \mathbf{D}_{12}^t & \mathbf{D}_{22} \end{bmatrix} \succeq 0.$$

Moreover, let

$$\mathbf{R} = \begin{bmatrix} \mathbf{R}_{11} & \mathbf{R}_{12} \\ \mathbf{R}_{12}^t & \mathbf{R}_{22} \end{bmatrix},$$

where

$$\mathbf{R}_{11} = \alpha\mathbf{I}_{r_1} + \beta\mathbf{1}_{r_1} \tag{55}$$
$$\mathbf{R}_{22} = \alpha\mathbf{I}_{r_2} + \beta\mathbf{1}_{r_2}$$
$$\mathbf{R}_{12} = \beta\mathbf{1}_{r_1 \times r_2},$$

such that

$$\alpha = 1 - \frac{2}{\pi}$$
$$\beta = \frac{2}{\pi} + \frac{1}{\pi d}.$$

Let

$$\boldsymbol{\Delta} = \mathbf{D} - \mathbf{R} = \begin{bmatrix} \boldsymbol{\Delta}_{11} & \boldsymbol{\Delta}_{12} \\ \boldsymbol{\Delta}_{12}^t & \boldsymbol{\Delta}_{22} \end{bmatrix}.$$

Note that to simplify notations, we make the dependency of these matrices to $d$, $r_1$ and $r_2$ implicit. Using Theorem 2.1 of reference [32], under the assumptions of the theorem, as $d, r_1 \to \infty$, we have $\|\boldsymbol{\Delta}_{11}\| \to 0$, $\|\boldsymbol{\Delta}_{22}\| \to 0$ and $\|\boldsymbol{\Delta}_{12}\| \to 0$ in probability. Moreover, we have

$$\mathbf{D}/\mathbf{D}_{11} = \mathbf{D}_{22} - \mathbf{D}_{12}^t\mathbf{D}_{11}^{-1}\mathbf{D}_{12} \tag{56}$$
$$= (\mathbf{R}_{22} + \boldsymbol{\Delta}_{22}) - (\mathbf{R}_{12} + \boldsymbol{\Delta}_{12})^t(\mathbf{R}_{11} + \boldsymbol{\Delta}_{11})^{-1}(\mathbf{R}_{12} + \boldsymbol{\Delta}_{12}).$$

Since $\lambda_{\min}(\mathbf{R}_{11}) = 1 - 2/\pi$, using Lemma 8, we have

$$(\mathbf{R}_{11} + \boldsymbol{\Delta}_{11})^{-1} = \mathbf{R}_{11}^{-1} + \mathbf{R}_{11}^{-1}\tilde{\boldsymbol{\Delta}}_{11}\mathbf{R}_{11}^{-1}, \tag{57}$$

where $\|\tilde{\boldsymbol{\Delta}}\| \to 0$ in probability. Using this equation in (56), we have

$$\mathbf{D}/\mathbf{D}_{11} = \mathbf{Z}_1 + \mathbf{Z}_2 \tag{58}$$

where

$$\mathbf{Z}_1 = \mathbf{R}_{22} - \mathbf{R}_{12}^t\mathbf{R}_{11}^{-1}\mathbf{R}_{12} \tag{59}$$

and

$$\mathbf{Z}_2 = \boldsymbol{\Delta}_{22} - \boldsymbol{\Delta}_{12}^t\mathbf{R}_{11}^{-1}\mathbf{R}_{12} - \boldsymbol{\Delta}_{12}^t\mathbf{R}_{11}^{-1}\boldsymbol{\Delta}_{12} \tag{60}$$
$$- \boldsymbol{\Delta}_{12}^t\mathbf{R}_{11}^{-1}\tilde{\boldsymbol{\Delta}}_{11}\mathbf{R}_{11}^{-1}\mathbf{R}_{12} - \boldsymbol{\Delta}_{12}^t\mathbf{R}_{11}^{-1}\tilde{\boldsymbol{\Delta}}_{11}\mathbf{R}_{11}^{-1}\boldsymbol{\Delta}_{12}$$
$$- \mathbf{R}_{12}^t\mathbf{R}_{11}^{-1}\boldsymbol{\Delta}_{12} - \mathbf{R}_{12}^t\mathbf{R}_{11}^{-1}\tilde{\boldsymbol{\Delta}}_{11}\mathbf{R}_{11}^{-1}\mathbf{R}_{12} - \mathbf{R}_{12}^t\mathbf{R}_{11}^{-1}\tilde{\boldsymbol{\Delta}}_{11}\mathbf{R}_{11}^{-1}\boldsymbol{\Delta}_{12}.$$

First, we show that as $d, r_1 \to \infty$, $\|\mathbf{Z}_2\| \to 0$ in probability. Note that using Lemma 9, we have

$$\mathbf{R}_{11}^{-1} = \frac{1}{\alpha}\mathbf{I}_{r_1} - \frac{\beta}{\alpha^2 + \alpha\beta r_1}\mathbf{1}_{r_1}. \tag{61}$$

Therefore, we have

$$\mathbf{1}_{r_2 \times r_1}\mathbf{R}_{11}^{-1} = \frac{1}{\alpha + \beta r_1}\mathbf{1}_{r_2 \times r_1}. \tag{62}$$

Thus, we have

$$\|\mathbf{1}_{r_2 \times r_1}\mathbf{R}_{11}^{-1}\| \le c_1 \tag{63}$$

for sufficiently large $r_1$. Similarly, we have

$$\|\mathbf{R}_{11}^{-1}\| \le c_2, \tag{64}$$

for sufficiently large $r_1$. Using (63) and (64) in (60), it is straightforward to show that as $d, r_1 \to \infty$, $\|\mathbf{Z}_2\| \to 0$ in probability.

Next, we characterize $\|\mathbf{Z}_1\|$. We have

$$\mathbf{Z}_1 = \alpha \mathbf{I}_{r_2} + \beta \mathbf{1}_{r_2} - \beta^2 \mathbf{1}_{r_2 \times r_1} \mathbf{R}_{11}^{-1} \mathbf{1}_{r_1 \times r_2} \tag{65}$$

$$= \alpha \mathbf{I}_{r_2} + \frac{\alpha \beta}{\alpha + \beta r_1} \mathbf{1}_{r_2}.$$

Therefore, we have

$$\|\mathbf{Z}_1\| = \alpha \left( 1 + \frac{\beta r_2}{\alpha + \beta r_1} \right) \tag{66}$$

$$= \left( 1 - \frac{2}{\pi} \right) \left( 1 + \left( 1 - \frac{\pi - 2}{\gamma + \pi - 2 + 2r_1} \right) \frac{r_2}{r_1} \right)$$

$$= \left( 1 - \frac{2}{\pi} \right) \left( 1 + \frac{r_2}{r_1} \right),$$

as $r_1 \to \infty$. This completes the proof.

## 10.15   Proof of Proposition MT-1

Since $\mathbf{q}^*$ is a vector in $\mathbb{R}^{r^*}$ whose components are non-negative, we can write

$$\mathbf{q}^* = \frac{\|\mathbf{q}^*\|_1}{r^*} \mathbf{1}_{r^* \times 1} + \mathbf{q}_2^*, \tag{67}$$

where $\mathbf{q}_2^*$ is orthogonal to the vector $\mathbf{1}_{r^* \times 1}$. Therefore, we have

$$L(\mathbf{W} = 0) = \frac{1}{4} \| \sum_{i=1}^{r^*} \mathbf{w}_i^* \|^2 + \frac{1}{4} (\mathbf{q}^*)^t \, \psi[\mathbf{K}_{\mathcal{L}^*}] \mathbf{q}^* \tag{68}$$

$$\geq \frac{1}{4} (\mathbf{q}^*)^t \left( (1 - \frac{2}{\pi}) \mathbf{I}_{r^*} + (\frac{2}{\pi} + \frac{1}{\pi d}) \mathbf{1}_{r^* \times r^*} \right) \mathbf{q}^*$$

$$= \frac{1}{4} (1 - \frac{2}{\pi}) \|\mathbf{q}^*\|^2 + \frac{1}{2\pi} \|\mathbf{q}^*\|_1^2$$

$$\geq \frac{1}{4} \|\mathbf{q}^*\|^2,$$

where the first step follows from Theorem MT-3, the second step follows from (69), the third step follows from (67) and the fact that $d \to \infty$, and the last step follows from the fact that $\|\mathbf{q}^*\|_1 \geq \|\mathbf{q}^*\|$. Using (68) in Theorem MT-6 completes the proof.

## 10.16   Proof of Lemma 2

To simplify notations, define

$$\mathbf{D} = \psi[\mathbf{K}] = \begin{bmatrix} \mathbf{D}_{11} & \mathbf{D}_{12} \\ \mathbf{D}_{12}^t & \mathbf{D}_{22} \end{bmatrix} \succeq 0$$

We also use $\mathbf{U}$ instead of $\mathbf{U}_{\mathcal{L}}$.

Let $\mathbf{w}_j^+ = \sum_{i: \mathbf{w}_i = \|\mathbf{w}_i\| \mathbf{u}_j} \|\mathbf{w}_i\|$ and $\mathbf{w}_j^- = \sum_{i: \mathbf{w}_i = -\|\mathbf{w}_i\| \mathbf{u}_j} \|\mathbf{w}_i\|$. Thus, we have

$$\mathbf{w}_j^+ - \mathbf{w}_j^- = s_j q_j.$$

Hence,

$$\sum_{i=1}^{k} \mathbf{w}_i = \sum_{j=1}^{r_1} \left( \mathbf{w}_j^+ - \mathbf{w}_j^- \right) \mathbf{u}_j = \sum_{j=1}^{r_1} s_j q_j \mathbf{u}_j = \mathbf{USq}.$$

Therefore, equation (8) implies that

$$\mathbf{SU}^t \left( \mathbf{USq} - \mathbf{w}_0 \right) + \mathbf{D}_{11} \mathbf{q} - \mathbf{D}_{12} \mathbf{q}^* = 0.$$

Thus,
$$\mathbf{q} = \left(\mathbf{SU}^t\mathbf{US} + \mathbf{D}_{11}\right)^\dagger \left(\mathbf{SU}^t\mathbf{w}_0 + \mathbf{D}_{12}\mathbf{q}^*\right)$$
and
$$-\mathbf{SU}^t\mathbf{z} = \mathbf{D}_{11}\mathbf{q} - \mathbf{D}_{12}\mathbf{q}^*$$
$$= \mathbf{D}_{11}\left(\mathbf{SU}^t\mathbf{US} + \mathbf{D}_{11}\right)^\dagger \mathbf{SU}^t\mathbf{w}_0 + \left(\mathbf{D}_{11}\left(\mathbf{SU}^t\mathbf{US} + \mathbf{D}_{11}\right)^\dagger - \mathbf{I}\right)\mathbf{D}_{12}\mathbf{q}^*.$$

Thus,
$$\mathbf{z} = -\left(\mathbf{USS}^t\mathbf{U}^t\right)^{-1}\mathbf{US}\left[\mathbf{D}_{11}\left(\mathbf{SU}^t\mathbf{US} + \mathbf{D}_{11}\right)^\dagger \mathbf{SU}^t\mathbf{w}_0 + \left(\mathbf{D}_{11}\left(\mathbf{SU}^t\mathbf{US} + \mathbf{D}_{11}\right)^\dagger - \mathbf{I}\right)\mathbf{D}_{12}\mathbf{q}^*\right].$$

### 10.17 Proof of Theorem 6

To simplify notations, define
$$\mathbf{D} = \psi[\mathbf{K}] = \begin{bmatrix} \mathbf{D}_{11} & \mathbf{D}_{12} \\ \mathbf{D}_{12}^t & \mathbf{D}_{22} \end{bmatrix} \succeq 0$$

We also use $\mathbf{U}$ instead of $\mathbf{U}_{\mathcal{L}}$.

Under assumptions 1, (11) simplifies to
$$\mathbf{z} = -\left(\mathbf{UU}^t\right)^{-1}\mathbf{UD}_{11}\left(\mathbf{D}_{11}\left(\mathbf{U}^t\mathbf{U} + \mathbf{D}_{11}\right)^{-1} - \mathbf{I}\right)\mathbf{D}_{12}\mathbf{q}^*.$$

Using the Woodbury matrix identity,
$$\left(\mathbf{D}_{11} + \mathbf{U}^t\mathbf{U}\right)^{-1} = \mathbf{D}_{11}^{-1} - \mathbf{D}_{11}^{-1}\mathbf{U}^t\left(\mathbf{I} + \mathbf{UD}_{11}^{-1}\mathbf{U}^t\right)^{-1}\mathbf{UD}_{11}^{-1}.$$

Hence,
$$\mathbf{z} = \left(\mathbf{I} + \mathbf{UD}_{11}^{-1}\mathbf{U}^t\right)^{-1}\mathbf{UD}_{11}^{-1}\mathbf{D}_{12}\mathbf{q}^*.$$

Therefore,
$$\left\langle \mathbf{z}, \left(\mathbf{I} + \mathbf{UD}_{11}^{-1}\mathbf{U}^t\right)\mathbf{z} \right\rangle = \left\langle \mathbf{q}^*, \mathbf{D}_{12}^t\mathbf{D}_{11}^{-1}\mathbf{U}^t\left(\mathbf{I} + \mathbf{UD}_{11}^{-1}\mathbf{U}^t\right)^{-1}\mathbf{UD}_{11}^{-1}\mathbf{D}_{12}\mathbf{q}^* \right\rangle.$$

Replacing this in (9), we get
$$L(\mathbf{W}) = \frac{1}{4}\left\langle \mathbf{q}^*, \left(\mathbf{D}_{22} - \mathbf{D}_{12}^t\mathbf{D}_{11}^{-1/2}\left(\mathbf{I} - \mathbf{D}_{11}^{-1/2}\mathbf{U}^t\left(\mathbf{I} + \mathbf{UD}_{11}^{-1}\mathbf{U}^t\right)^{-1}\mathbf{UD}_{11}^{-1}\right)\mathbf{D}_{11}^{-1/2}\mathbf{D}_{12}\right)\mathbf{q}^* \right\rangle.$$

Note that we can write
$$\mathbf{D}_{22} - \mathbf{D}_{12}^t\mathbf{D}_{11}^{-1/2}\left(\mathbf{I} - \mathbf{D}_{11}^{-1/2}\mathbf{U}^t\left(\mathbf{I} + \mathbf{UD}_{11}^{-1}\mathbf{U}^t\right)^{-1}\mathbf{UD}_{11}^{-1}\right)\mathbf{D}_{11}^{-1/2}\mathbf{D}_{12} = \widetilde{\mathbf{D}}/\mathbf{D}_{11},$$

where
$$\widetilde{\mathbf{D}} = \begin{bmatrix} \widetilde{\mathbf{D}}_{11} & \mathbf{D}_{12} \\ \mathbf{D}_{12}^t & \mathbf{D}_{22} \end{bmatrix},$$
$$\widetilde{\mathbf{D}}_{11} = \mathbf{D}_{11}^{1/2}\left(\mathbf{I} - \mathbf{D}_{11}^{-1/2}\mathbf{U}^t\left(\mathbf{I} + \mathbf{UD}_{11}^{-1}\mathbf{U}^t\right)^{-1}\mathbf{UD}_{11}^{-1}\right)^{-1}\mathbf{D}_{11}^{1/2}.$$

Using the Woodbury matrix identity one more time leads to
$$\left(\mathbf{I} - \mathbf{D}_{11}^{-1/2}\mathbf{U}^t\left(\mathbf{I} + \mathbf{UD}_{11}^{-1}\mathbf{U}^t\right)^{-1}\mathbf{UD}_{11}^{-1}\right)^{-1} = \mathbf{I} - \mathbf{D}_{11}^{-1/2}\mathbf{U}^t\left(-\mathbf{I} - \mathbf{UD}_{11}^{-1}\mathbf{U}^t + \mathbf{UD}_{11}^t\mathbf{U}^t\right)\mathbf{UD}_{11}^{-1/2}$$
$$= \mathbf{I} + \mathbf{D}_{11}^{-1/2}\mathbf{U}^t\mathbf{UD}_{11}^{-1/2}.$$

Thus,
$$\widetilde{\mathbf{D}}_{11} = \mathbf{D}_{11} + \mathbf{U}^t\mathbf{U}, \quad \widetilde{\mathbf{D}} = \begin{bmatrix} \mathbf{D}_{11} + \mathbf{U}^t\mathbf{U} & \mathbf{D}_{12} \\ \mathbf{D}_{12}^t & \mathbf{D}_{22} \end{bmatrix},$$

and
$$L(\mathbf{W}) = \frac{1}{4}\left\langle \mathbf{q}^*, \left(\widetilde{\mathbf{D}}/\mathbf{D}_{22}\right)\mathbf{q}^* \right\rangle.$$

This completes the proof.

## 10.18 Proof of Theorem 7

To simplify notations, we use $\mathbf{U}$ instead of the $\mathbf{U}_{\mathcal{L}}$. Moreover, we define

$$\psi[\mathbf{K}] = \begin{bmatrix} \mathbf{D}_{11} & \mathbf{D}_{12} \\ \mathbf{D}_{12}^t & \mathbf{D}_{22} \end{bmatrix},$$

and

$$\widetilde{\mathbf{D}}_{11} = \mathbf{D}_{11} + \mathbf{U}^t\mathbf{U}, \quad \widetilde{\mathbf{D}} = \begin{bmatrix} \mathbf{D}_{11} + \mathbf{U}^t\mathbf{U} & \mathbf{D}_{12} \\ \mathbf{D}_{12}^t & \mathbf{D}_{22} \end{bmatrix}.$$

Moreover, let

$$\mathbf{R} = \begin{bmatrix} \mathbf{R}_{11} & \mathbf{R}_{12} \\ \mathbf{R}_{12}^t & \mathbf{R}_{22} \end{bmatrix},$$

where

$$\begin{align} \mathbf{R}_{11} &= \alpha\mathbf{I}_{r_1} + \beta\mathbf{1}_{r_1} + \mathbf{U}^t\mathbf{U} \tag{69} \\ \mathbf{R}_{22} &= \alpha\mathbf{I}_{r_2} + \beta\mathbf{1}_{r_2} \\ \mathbf{R}_{12} &= \beta\mathbf{1}_{r_1 \times r_2}, \end{align}$$

such that

$$\alpha = 1 - \frac{2}{\pi}$$
$$\beta = \frac{2}{\pi} + \frac{1}{\pi d}.$$

Let

$$\mathbf{\Delta} = \mathbf{R} - \widetilde{\mathbf{D}} = \begin{bmatrix} \mathbf{\Delta}_{11} & \mathbf{\Delta}_{12} \\ \mathbf{\Delta}_{12}^t & \mathbf{\Delta}_{22} \end{bmatrix}.$$

Using the result of Theorem 6, we have

$$L(\mathbf{W}) = \frac{1}{4}\left\langle \mathbf{q}^*, \left(\widetilde{\mathbf{D}}/\mathbf{D}_{22}\right)\mathbf{q}^* \right\rangle \le \frac{1}{4}\left\| \left(\widetilde{\mathbf{D}}/\mathbf{D}_{22}\right) \right\|_2 \|\mathbf{q}^*\|_2^2.$$

Similar to the proof of Theorem MT-6, the $\mathbf{\Delta}$ matrix and the $1/d$ term of $\beta$ have negligible effects in the asymptotic regime. Hence, it is sufficient to bound $\|\mathbf{R}/\mathbf{R}_{11}\|_2$. We have

$$\mathbf{R}/\mathbf{R}_{11} = (\beta\mathbf{1}_{r_2} + \alpha\mathbf{I}_{r_2}) - \beta^2\left(\beta\mathbf{1}_{r_1} + \alpha\mathbf{I}_{r_1} + \mathbf{U}^t\mathbf{U}\right)^{-1}\mathbf{1}_{r_1 \times r_2}. \tag{70}$$

Note that if $\mathbf{u} \in \mathbb{R}^{r_2}$ where $\|\mathbf{u}\| = 1$ and $<\mathbf{u}, \mathbf{1}>= 0$, we have

$$(\mathbf{R}/\mathbf{R}_{11})\,\mathbf{u} = \alpha\mathbf{u} \tag{71}$$

which leads to $\|(\mathbf{R}/\mathbf{R}_{11})\mathbf{u}\| = \alpha$ and $<\mathbf{u}, (\mathbf{R}/\mathbf{R}_{11}\mathbf{u}) >= \alpha$. Moreover, we have

$$\lim_{d\to\infty} \frac{1}{r_2}\langle \mathbf{1}, (\mathbf{R}/\mathbf{R}_{11})\mathbf{1}\rangle = \lim_{d\to\infty} \frac{1}{r_2}\langle \mathbf{1}_{r_2}, (\mathbf{R}/\mathbf{R}_{11})\rangle. \tag{72}$$

Using the Woodbury matrix identity and Lemma 9, we have

$$\left(\frac{2}{\pi}\mathbf{1}_{r_1} + \alpha\mathbf{I}_{r_1} + \mathbf{U}^t\mathbf{U}\right)^{-1} = \left(\frac{2}{\pi} + \alpha\mathbf{I}_{r_1}\right)^{-1} \tag{73}$$

$$- \left(\frac{2}{\pi} + \alpha\mathbf{I}_{r_1}\right)^{-1}\mathbf{U}^t\left(\mathbf{I} + \mathbf{U}\left(\frac{2}{\pi}\mathbf{1}_{r_1} + \alpha\mathbf{I}_{r_1}\right)^{-1}\mathbf{U}^t\right)^{-1}\mathbf{U}\left(\frac{2}{\pi} + \alpha\mathbf{I}_{r_1}\right)^{-1}$$

$$= \left(\frac{1}{\alpha}\mathbf{I}_{r_1} - \frac{2}{\alpha(\pi\alpha + 2r_1)}\mathbf{1}_{r_1}\right)$$

$$- \left(\frac{1}{\alpha}\mathbf{I}_{r_1} - \frac{2}{\alpha(\pi\alpha + 2r_1)}\mathbf{1}_{r_1}\right)\mathbf{U}^t\left(\mathbf{I} + \mathbf{U}\left(\frac{1}{\alpha}\mathbf{I}_{r_1} - \frac{2}{\alpha(\pi\alpha + 2r_1)}\mathbf{1}_{r_1}\right)\mathbf{U}^t\right)^{-1}\mathbf{U}\left(\frac{1}{\alpha}\mathbf{I}_{r_1} - \frac{2}{\alpha(\pi\alpha + 2r_1)}\mathbf{1}_{r_1}\right).$$

Letting

$$\mathbf{A} := \mathbf{U}^t \left( \mathbf{I} + \mathbf{U} \left( \frac{1}{\alpha} \mathbf{I}_{r_1} - \frac{2}{\alpha(\pi\alpha + 2r_1)} \mathbf{1}_{r_1} \right) \mathbf{U}^t \right)^{-1} \mathbf{U}, \tag{74}$$

we have

$$\frac{4}{\pi^2} \mathbf{1}_{r_2 \times r_1} \left( \frac{2}{\pi} \mathbf{1}_{r_1} + \alpha \mathbf{I}_{r_1} + \mathbf{U}^t \mathbf{U} \right)^{-1} \mathbf{1}_{r_1 \times r_2} = \frac{4}{\pi^2} \left( \frac{r_1}{2/\pi r_1 + \alpha} \mathbf{1}_{r_2} - \frac{1}{(2/\pi r_1 + \alpha)^2} \langle \mathbf{1}_{r_1}, \mathbf{A} \rangle \mathbf{1}_{r_2} \right). \tag{75}$$

Therefore, using (70), we have

$$\frac{1}{r_2} \langle \mathbf{1}, \mathbf{R}/\mathbf{R}_{11} \rangle = \frac{2r_2}{\pi} + \alpha - \frac{(4/\pi^2) r_1 r_2}{2/\pi r_1 + \alpha} + \frac{(4/\pi^2) \langle \mathbf{1}_{r_1}, \mathbf{A} \rangle r_2}{(2r_1/\pi + \alpha)^2}. \tag{76}$$

Therefore, we have

$$\lim_{d \to \infty} \frac{1}{r_2} \langle \mathbf{1}, \mathbf{R}/\mathbf{R}_{11} \rangle = \alpha + \langle \mathbf{1}_{r_1}, \mathbf{A} \rangle \frac{r_2}{r_1^2}. \tag{77}$$

On the other hand, since the matrix $1/\alpha \mathbf{I} - 2/(\alpha(\pi\alpha + 2r_1))\mathbf{1}_{r_1}$ is positive semidefinite, we have

$$\langle \mathbf{1}_{r_1}, \mathbf{A} \rangle \leq \langle \mathbf{1}_{r_1}, \mathbf{U}^t \mathbf{U} \rangle = \|\mathbf{U}\mathbf{1}_{r_1}\|^2. \tag{78}$$

Since columns of $\mathbf{U}$ are randomly generated (e.g., using a Gaussian distribution), we have $\|\mathbf{U}\| \leq 1 + \sqrt{\gamma} + \mu$ with probability $1 - 2\exp(-\mu^2 d)$. Thus, $\|\mathbf{U}\mathbf{1}\|^2 \leq r(1 + \sqrt{\gamma} + \mu)^2$ with probability $1 - 2\exp(-\mu^2 d)$. Thus, with high probability,

$$\lim_{d \to \infty} \frac{1}{r_2} \langle \mathbf{1}, \mathbf{R}/\mathbf{R}_{11} \rangle \leq 1 - \frac{2}{\pi} + (1 + \sqrt{\gamma} + \mu)^2 \frac{r_2}{r_1}. \tag{79}$$

This along with (71) lead to

$$\|\mathbf{R}/\mathbf{R}_{11}\| \leq 1 - \frac{2}{\pi} + (1 + \sqrt{\gamma} + \mu)^2 \frac{r_2}{r_1} \tag{80}$$

with probability $1 - 2\exp(-\mu^2 d)$. Replacing this in (12) completes the proof.

### 10.19 Proof of Lemma 3

We consider four different cases for signs of $\langle \mathbf{w}_1, \mathbf{x} \rangle$, $\langle \mathbf{w}_2, \mathbf{x} \rangle$.

1. $\langle \mathbf{w}_1, \mathbf{x} \rangle \leq 0$, $\langle \mathbf{w}_2, \mathbf{x} \rangle \leq 0$: In this case, $\phi(\langle \mathbf{w}_1, \mathbf{x} \rangle) = \phi(\langle \mathbf{w}_2, \mathbf{x} \rangle) = 0$. Hence, the lemma statement is trivial.

2. $\langle \mathbf{w}_1, \mathbf{x} \rangle \geq 0$, $\langle \mathbf{w}_2, \mathbf{x} \rangle \geq 0$: We have

   $$\phi(\langle \mathbf{w}_1, \mathbf{x} \rangle) - \phi(\langle \mathbf{w}_2, \mathbf{x} \rangle) = \langle \mathbf{w}_1, \mathbf{x} \rangle - \langle \mathbf{w}_2, \mathbf{x} \rangle = \langle \mathbf{w}_1 - \mathbf{w}_2, \mathbf{x} \rangle \leq \|\mathbf{w}_1 - \mathbf{w}_2\|_2 \|\mathbf{x}\|_2.$$

3. $\langle \mathbf{w}_1, \mathbf{x} \rangle \geq 0$, $\langle \mathbf{w}_2, \mathbf{x} \rangle \leq 0$: In this case we have

   $$\phi(\langle \mathbf{w}_1, \mathbf{x} \rangle) - \phi(\langle \mathbf{w}_2, \mathbf{x} \rangle) = \langle \mathbf{w}_1, \mathbf{x} \rangle = \langle \mathbf{w}_1 - \mathbf{w}_2, \mathbf{x} \rangle + \langle \mathbf{w}_2, \mathbf{x} \rangle \leq \langle \mathbf{w}_1 - \mathbf{w}_2, \mathbf{x} \rangle \leq \|\mathbf{w}_1 - \mathbf{w}_2\|_2 \|\mathbf{x}\|_2.$$

4. $\langle \mathbf{w}_1, \mathbf{x} \rangle \leq 0$, $\langle \mathbf{w}_2, \mathbf{x} \rangle \geq 0$: After switching the roles of $\mathbf{w}_1, \mathbf{w}_2$, the proof is the same as it was in case (3).

Therefore, the lemma statement holds in all four cases for signs of $\langle \mathbf{w}_1, \mathbf{x} \rangle$, $\langle \mathbf{w}_2, \mathbf{x} \rangle$. This completes the proof.

### 10.20 Proof of Lemma 4

We use the result of Lemma 5.2 in [33]. Let $|\mathcal{U}|$ be an $\epsilon$-net of $H^{n-1}$, an arbitrary unit hemisphere in $n$-dimensions, where

$$\epsilon = \sqrt{2 - 2\cos\delta}.$$

Using Lemma 5.2 in [33],

$$|\mathcal{U}| \leq \frac{1}{2}\left(1 + \frac{\sqrt{2}}{\sqrt{1-\cos\delta}}\right)^n.$$

Now we show that $\mathcal{U}$ is an angular $\delta$-net of $S^{n-1}$. Let $\mathbf{v} \in \mathbb{R}^n$ be an arbitrary vector in $S^{n-1}$. Note that $\mathcal{U} \cup \mathcal{U}^-$ is an $\epsilon$-net for the unit sphere $S^{n-1}$. Hence, there exists a vector $\mathbf{u} \in \mathcal{U} \cup \mathcal{U}^-$, such that

$$\|\mathbf{u}-\mathbf{v}\|_2^2 \leq \epsilon^2 = 2 - 2\cos\delta. \tag{81}$$

Thus,

$$\|\mathbf{u}\|_2^2 + \|\mathbf{v}\|_2^2 - 2\|\mathbf{u}\|\|\mathbf{v}\|\cos\theta_{\mathbf{u},\mathbf{v}} = 2 - 2\cos\theta_{\mathbf{u},\mathbf{v}} \leq 2 - 2\cos\delta.$$

Therefore,

$$\cos\theta_{\mathbf{u},\mathbf{v}} \geq \cos\delta \Rightarrow \theta_{\mathbf{u},\mathbf{v}} \leq \delta.$$

Hence, for every vector $\mathbf{v} \in S^{n-1}$, there exists $\mathbf{u} \in \mathcal{U} \cup \mathcal{U}^-$, such that

$$\theta_{\mathbf{u},\mathbf{v}} \leq \delta.$$

This completes the proof.

### 10.21   Proof of Theorem 8

Let $f^*(\mathbf{x}) = h(\mathbf{x}; \mathbf{w}_1^*, \mathbf{w}_2^*, \ldots, \mathbf{w}_k^*)$, for a set of weights $\mathbf{w}_i^* \in \mathcal{W}$, be an arbitrary member of $\mathcal{F}$. Since $\mathcal{U}$ is an angular $\delta$-net of $\mathcal{W}$, for $i = 1, 2, \ldots, k$, we can take $\tilde{\mathbf{u}}_i \in \mathcal{U} \cup \mathcal{U}^-$ such that $\theta_{\tilde{\mathbf{u}}_i, \mathbf{w}_i^*} \leq \delta$. For $i = 1, 2, \ldots, k$, take $\tilde{\mathbf{w}}_i \in \mathcal{W}_\mathcal{U}$ as

$$\tilde{\mathbf{w}}_i = \frac{\|\mathbf{w}_i^*\|}{\|\tilde{\mathbf{u}}_i\|}\tilde{\mathbf{u}}_i.$$

Note that we have

$$\|\mathbf{w}_i^* - \tilde{\mathbf{w}}_i\|_2^2 = \|\mathbf{w}_i^*\|_2^2 + \|\tilde{\mathbf{w}}_i\|_2^2 - 2\|\tilde{\mathbf{w}}_i\|_2\|\mathbf{w}_i^*\|_2 \cos\theta_{\tilde{\mathbf{u}}_i, \mathbf{w}_i^*}$$
$$= 2\|\mathbf{w}_i^*\|_2^2(1 - \cos\theta_{\tilde{\mathbf{u}}_i, \mathbf{w}_i^*}) \leq 2\|\mathbf{w}_i^*\|_2^2(1 - \cos\delta). \tag{82}$$

Taking $\tilde{f}(\mathbf{x}) = h(\mathbf{x}; \tilde{\mathbf{w}}_1, \tilde{\mathbf{w}}_2, \ldots, \tilde{\mathbf{w}}_k) \in \mathcal{F}_\mathcal{L}$, we have

$$\min_{\hat{f} \in \mathcal{F}_\mathcal{L}} \mathbb{E}\left|f(\mathbf{x}) - \hat{f}(\mathbf{x})\right| \leq \mathbb{E}|f(\mathbf{x}) - \tilde{f}(\mathbf{x})| \leq \mathbb{E}|h(\mathbf{x}; \mathbf{w}_1^*, \mathbf{w}_2^*, \ldots, \mathbf{w}_k^*) - h(\mathbf{x}; \tilde{\mathbf{w}}_1, \tilde{\mathbf{w}}_2, \ldots, \tilde{\mathbf{w}}_k^*)|$$

$$\leq \mathbb{E}\left|\sum_{i=1}^{k}\phi\left(\langle\mathbf{w}_i^*, \mathbf{x}\rangle\right) - \sum_{i=1}^{k}\phi\left(\langle\tilde{\mathbf{w}}_i, \mathbf{x}\rangle\right)\right|$$

$$\leq \mathbb{E}\sum_{i=1}^{k}\left|\phi\left(\langle\mathbf{w}_i^*, \mathbf{x}\rangle\right) - \phi\left(\langle\tilde{\mathbf{w}}_i, \mathbf{x}\rangle\right)\right|.$$

Using Lemma 3, we get

$$\min_{\hat{f} \in \mathcal{F}_\mathcal{L}} \mathbb{E}\left|f(\mathbf{x}) - \hat{f}(\mathbf{x})\right| \leq \left(\sum_{i=1}^{k}\|\mathbf{w}_i^* - \tilde{\mathbf{w}}_i\|_2\right)\mathbb{E}\|\mathbf{x}\|_2 = \sqrt{d}\sum_{i=1}^{k}\|\mathbf{w}_i^* - \tilde{\mathbf{w}}_i\|_2$$

Hence, by (82)

$$\min_{\hat{f} \in \mathcal{F}_\mathcal{L}} \mathbb{E}\left|f(\mathbf{x}) - \hat{f}(\mathbf{x})\right| \leq \sqrt{2d(1-\cos\delta)}\sum_{i=1}^{k}\|\mathbf{w}_i^*\|_2 \leq kM\sqrt{2d(1-\cos\delta)}. \tag{83}$$

Thus,

$$\mathcal{R}\left(\mathcal{F}_\mathcal{L}, \mathcal{F}\right) = \max_{f \in \mathcal{F}}\min_{\hat{f} \in \mathcal{F}_\mathcal{V}} \mathbb{E}\left|f(\mathbf{x}) - \hat{f}(\mathbf{x})\right| \leq kM\sqrt{2d(1-\cos\delta)}.$$

## Footnotes

[1] These definitions match with definitions of $\mathcal{G}$ and $g(.)$ for a general PNN.