[Reviews · NeurIPS 2018]

Reviewer 1



This paper presents a new type of neural network where the weights are constrained to lie over a fixed set of lines (called PPN lines). Although this reduces the capacity of the network, it still yields a high-capacity network and more importantly, it “simplifies” the energy landscape making it easier for an optimizer to find the optimum weights. I do have some comments, mostly regarding the relation to previous work or some technical part in the analysis. I will consider raising my scores if these questions are addressed in the rebuttal. 1) Comparison to existing approaches I think the approach presented by the authors is interesting as it offers some improvements over past approaches. As pointed out by the authors, most existing work assume the number of neurons is less than the dimensions. However, there are exceptions and I think it’s worth pointing this out. For instance, Soltanolkotabi et al. (2017) already focused on the (more reasonable) over-parametrized regime where n < O(d). 2) Comparison to other architectures such as resnet or densenet The authors essentially suggest a way to reduce the capacity of the network. There are already many ways to modify the architecture of a neural network to achieve similar effects such as using residual or skip connections. See e.g. https://arxiv.org/pdf/1712.09913.pdf for a discussion of the effect of these connections on the landscape of a neural network. 3) Characterization global minima Vidal et al. (https://arxiv.org/pdf/1712.04741.pdf) gave conditions for global optimality of a neural network (related to the number of zero weights). This seems to relate to the conditions given in Theorem 2, can you comment on this? 4) Choice of PNN lines How important is the choice of PNN lines on which your project the weights? For instance, would you get similar guarantees with random projections? Would this simply affect the eigenvalues of the kernel function? 5) Gaussian input “We analyze population risk landscapes of two-layer PNNs with jointly Gaussian inputs” It’s not clear to me where in the analysis you require the Gaussian input? Can you please comment on this? 6) Choice of activation function The analysis relies on the a rectified linear unit (ReLU) activation function. How important is this choice? Can you generalize it to other types of activation functions? 7) Extension to >2 hidden layers Can the authors comment on extending their approach to more than 2 layers? 8) Proof of Theorem 2 for characterization of global optima You seem to only consider the fact that the norm of the gradient should be zero. Aren’t you only consider first-order critical points? What about the eigenvalues of the Hessian? In theorem 3, you said the Hessian is positive semidefinite which does not rule out zero-eigenvalues (i.e. degenerate points). 9) Global optimizer The authors seem to assume the existence of one global minimizer in all their theorems, even in the case where every local optimizer it not a global optimizer. For instance, take Theorem 1 in the case where the conditions of Theorem 2 are not satisfied. 10) Effect of variables d (input dimension) and r (number of PNN lines) As pointed out by the authors, Theorem 5 does not characterize the rate of the decrease as a function of d (and r). I wonder if the authors have conducted some experiments to see the empirical effect of d and r. From a theory point of view, only the regime d->infty is analyzed in the paper. From an practitioner point of view, I think it would really be interesting to understand how large r should be to reach a good level of performance. 11) MNIST experiments: The authors only use a subset of MNIST (they only consider images of 1’s and 2’s). Can you elaborate on this choice? Where is the difficulty in extending this to >2 classes? 12) Appendix The numbering scheme used in the appendix is different from the main paper, sometime making it difficult to find the proof of a theorem. a) For instance, where is the proof of Theorem 1 in the appendix? The closest seems to be theorem 4 where the form of the Kernel C is much simpler that given in the main paper. Can you explain how you derive the form of the kernel giving in Theorem 1 in the main paper? b) Similar remarks for other theorems. I think you should consider re-organizing the supplementary material to make it easier for the reader to find the proofs of the theorems cited in the paper.

Reviewer 2



The paper proposes a variant of 2-layer neural networks called porcupine networks which are obtained by defining an alternative formulation (objective function) by constraining the weight vectors to lie on a favorable landscape. The advantage of this being that all local optima are also global. The idea is appealing to me theoretically as a proof of concept, however I would like to see some empirical evaluation (on synthetic datasets) using some simple neural nets. The authors have delved deeper into the theoretical guarantees and that is a plus of this work. I like the fact that constraining the non-convex optimization landscape can help in obtaining global optimum solutions. I would like to see the authors present a discussion of the true merits (or selling points) of using PNNs on real-world datasets and applications. Are they attractive from a memory storage point of view? Also, can the authors shed some light on whether PNNs be extended to other types of neural networks - for e.g. sequential data (RNNs, LSTMs)?

Reviewer 3



Summary: The authors present an angular discretization of a neural network with desirable properties that should also hold for the neural network if the discretization error is minimal. The proposed discretization is an attempt to provide a new tool to the toolkit of theorical analysis of deep neural networks. Quality: The experiment section is very short and it is not followed by a discussion, thus results analysis is not developed enough. The interpolation made between the PNNs properties and unconstrained neural networks based on the low approximation error of PNNs is not grounded on any proof. Clarity: The introduction is too long and contains too many details. Second and fourth paragraphs should be moved in a related work section. Paragraphs 3 and 6 contains too many details about the PNN model. This makes the reading difficult and the storyline becomes unclear. The idea is somehow simple add very easy to understand when presented as an 'angular discretization of functions'. This intuitive interpretation should be given in the introduction in a concise way. The base theory presented is clear but could also be more concise. It is not clear for example why they present a risk as the MSE. Is it necessary? If not then this kind of detail should be skipped. The definitions of matched and mismatched PNNs should be briefly reexplained in beginning of section 4 and 5. The proporties mentioned at line 124 should be at least enumerated in the core text. This sounds too important to be totally relayed to SM. The purpose of sections 4, 5 and 6 is difficult to understand. We end up with a bound on the approximation error based on the size of the PNN and the UNN, but it is not clear why would this information be helpful. Is the goal to infer correspondence of PNN's properties to UNNs? If yes, then the bound on the approximation error does not give any information about how accurate is this interpolation. At best we can only assume that the validity of the interpolation is inversely propertional to the approximation error. Originality: The discretization scheme presented seems remanescent of projection methods such as the one described in [1]. Maybe a comparision with such random projection methods should be given in this work. Significance: In my opinion this work would be hardly reusable unless a stronger proof is given for the correspondence of PNN and UNN properties.